# Single-cell analysis of developing and azoospermia human testicles reveals central role of Sertoli cells

LiangYu Zhao[1,2,6], ChenCheng Yao[1,2,6], XiaoYu Xing[3,6], Tao Jing[1,4,6], Peng Li[1], ZiJue Zhu[1], Chao Yang[1], Jing Zhai[1], RuHui Tian[1], HuiXing Chen[1], JiaQiang Luo[1], NaChuan Liu[1], ZhiWen Deng[2], XiaoHan Lin[2], Na Li[2], Jing Fang[2,5], Jie Sun[3✉], ChenChen Wang[5✉], Zhi Zhou[2✉] & Zheng Li [1✉]

Clinical efficacy of treatments against non-obstructive azoospermia (NOA), which affects 1% of men, are currently limited by the incomplete understanding of NOA pathogenesis and normal spermatogenic microenvironment. Here, we profile >80,000 human testicular single-cell transcriptomes from 10 healthy donors spanning the range from infant to adult and 7 NOA patients. We show that Sertoli cells, which form the scaffold in the testicular micro-environment, are severely damaged in NOA patients and identify the roadmap of Sertoli cell maturation. Notably, Sertoli cells of patients with congenital causes (Klinefelter syndrome and Y chromosome microdeletions) are mature, but exhibit abnormal immune responses, while the cells in idiopathic NOA (iNOA) are physiologically immature. Furthermore, we find that inhibition of Wnt signaling promotes the maturation of Sertoli cells from iNOA patients, allowing these cells to regain their ability to support germ cell survival. We provide a novel perspective on the development of diagnostic methods and therapeutic targets for NOA.

[1] Department of Andrology, the Center for Men's Health, Urologic Medical Center, Shanghai Key Laboratory of Reproductive Medicine, Shanghai General Hospital, Shanghai Jiao Tong University School of Medicine, Shanghai 200080, China. [2] School of Life Science and Technology, ShanghaiTech University, Shanghai 201210, China. [3] Department of Urology, Shanghai Children's Medical Center, School of Medicine, Shanghai Jiao Tong University, Shanghai 200120, China. [4] Department of Andrology, the Affiliated Hospital of Qingdao University, Qingdao 266000 Shandong, China. [5] Shanghai Advanced Research Institute, Stem Cell and Reproductive Biology Laboratory, Chinese Academy of Sciences, Shanghai 201210, China. [6]These authors contributed equally: LiangYu Zhao, ChenCheng Yao, XiaoYu Xing, Tao Jing. ✉email: sunjie@scmc.com.cn; wangcc@sari.ac.cn; zhouzhi@shanghaitech.edu.cn; lizhengboshi@sjtu.edu.cn

Loss of fertility can be devastating to a patient. Unfortunately, 15% of couples in the world are suffering from infertility[1]. In contrast to female infertility, which can often be treated by hormone therapy to stimulate oocyte production, male infertility caused by spermatogenesis abnormalities are much more difficult to treat[2]. Non-obstructive azoospermia (NOA) is the most serious form of male factor infertility, occurring in 10–15% of infertile men[1,3]. However, only a small percentage is caused by congenital factors such as Klinefelter syndrome (KS) and Y chromosome AZF region microdeletion (AZF_Del). Most of the remaining cases are due to unknown causes, also known as idiopathic NOA (iNOA), accounting for over 70%[3]. Although the final morphological feature in iNOA testis is that there are no or few spermatogenic cells, sperm can be found in 10% of iNOA patients by surgery[3]. Notably, the degree of testicular development may predict adverse patient outcomes. Hence, effective etiology analysis and therapies for patients with NOA are urgently required.

Spermatogenesis depends on the full maturation of the somatic microenvironment. It is a complicated process that occurs after birth and is completed after puberty. An altered microenvironment, which might affect fertility, has been observed in NOA patients[4,5]. However, our understanding of cells in the somatic microenvironment remains limited. Although most studies divided Sertoli cell development after birth into two stages: the immature and mature, it is largely unknown whether there exists any intermediate or transition states. Maturation of spermatogenic microenvironment is regulated by the dramatic changes in hormone levels during puberty[6]. Little is known about this maturation process, Sertoli cell heterogeneity, and interaction with other cell types at the molecular level. The same is true for other testicular cell populations, hampering our understanding of the pathogenesis of spermatogenic disorders.

The rapid development of single-cell RNA sequencing allows us to investigate individual cell populations in testis at high resolution. Most previous studies have focused on germ cell spermatogenesis itself[7,8]. A recent study of human testicular single-cell transcriptomes in puberty has revealed developmental changes in somatic cells[9]. However, we still do not know the major differentiation signals, metabolic characteristics, and cell–cell interactions of these cells at various developmental stages. Here, we profiled 88,723 individual testicular cells from 10 healthy subjects of various ages and 7 patients with one of the three most common types of NOA. Focusing on the somatic cell dataset, we identified three independent stages during Sertoli cell maturation, uncovered the pathological changes and maturation disorders in different types of NOA Sertoli cells. In addition, we obtained the evidence that the Wnt signaling pathway regulates the maturation of Sertoli cells in both normal and NOA patients. Collectively, our results provide in-depth insight into the maturation of the spermatogenic microenvironment and the mechanisms underlying pathogenesis, thereby offering new targets for NOA treatment strategies.

## Results

### Overview of the hierarchies of multiple cell populations in healthy and NOA human testes

To characterize the baseline cellular diversity of testicles during human developing and under pathological state, we profiled cells from 10 donors with normal spermatogenesis (aged 2–31 years) and 7 NOA patients (KS (3 cases), AZFa_Del (1 case), and iNOA (3 cases)) (Fig. 1a, b, and Supplementary Fig. 1a). The sex hormone levels of 10 healthy donors were within the normal range for their respective ages[10] (Supplementary Fig. 1b). However, hormonal parameters of three types of NOA patients showed hypergonadotropin in all NOA patients, and high estradiol level in KS and iNOA patients

(Supplementary Fig. 1c). The PAS/hematoxylin staining results showed that all 10 physiological testes had a normal morphology with their ages (Supplementary Fig. 1d). In addition, to further identify the spermatogenic stages in each sample, we divided germ cell clusters into 14 sub-clusters according to classical and recently reported markers (Supplementary Fig. 1e). Some of these markers, including UTF1 (spermatogonial stem cells, subset 1), GFRA1 (spermatogonial stem cells, subset 2), PLZF (spermatogonial stem cells, subset 2), SMS (highly expressed in spermatogonial stem cells, subset 3), c-KIT (differentiating and differentiated spermatogonia), TKTL1 (expressed from spermatogonial stem cells to differentiated spermatogonia), SYCP3/γH2AX (spermatocytes), and PNA (spermatids/spermatozoa) were checked by immunohistochemical (IHC) stain (Supplementary Fig. 1e)[8,11,12]. These spermatogenic stages accorded with the ages of the healthy donors (Supplementary Fig. 1f). Uniform manifold approximation and projection (UMAP) analysis of these cells showed that the heterogeneity in different age groups was obvious, but all 5 healthy adult samples had a good repeatability (Fig. 1b and Supplementary Fig. 2d). On average, 9359 reads (UMIs), 2719 expressed genes and 3.7% mitochondrial genes were detected per cell (Supplementary Fig. 2a).

A total of nine cell clusters were identified in the whole cell population of 10 healthy subjects based on the expression of known cell type-specific markers (Supplementary Fig. 2g). These nine clusters could be further divided into two parts, germ cells and microenvironment somatic cells (Fig. 1c). The latter included endotheliocytes, macrophages, vascular smooth muscle cells (VSM_cells), peritubular myoid (PTM) cells, Sertoli cells, and Leydig cells. Total 1101 differentially expressed genes (DEGs) with a fold change of log$_2$ transformed UMI > 1 of each cluster were identified (Supplementary Data 1). The results of our gene ontology (GO) analysis of these DEGs were consistent with our understanding of the biological processes of these cell types (Supplementary Fig. 2e, f).

To gain insight into the cellular differences between testes form healthy subjects and NOA patients, we re-clustered testicular somatic cells from 10 healthy donors with AZFa_Del, KS, and iNOA (Fig. 1d–f). Sertoli cells between normal adult and three types of NOA showed the greatest dissimilarity than other somatic cell clusters according to the result of Jaccard and Bray-curtis distance (Fig. 1g). Furthermore, to eliminate the effects of germ cells, we also compared the dissimilarity of Sertoli cells within three types of NOA. Sertoli cells also showed the greatest dissimilarity among somatic cells (Supplementary Fig. 1g). Considering the fact that Sertoli cells, which are located around and are in direct contact with germ cells, function as scaffolds and "nurse" cells in the spermatogenic microenvironment[13], our results suggest that the changes of somatic microenvironment in NOA patients are mainly manifest in Sertoli cells. So further research into Sertoli cells could help to understand the defects of somatic microenvironment in NOA.

### Identification of three stages and heterogeneity of human Sertoli cells during maturation

To explore the heterogeneity of Sertoli cells during normal development, we re-clustered the Sertoli cells and identified three subpopulations in 10 healthy subjects (Fig. 2a, b). To analyze the origin and maturation process of Sertoli cells, we performed pseudotime trajectory analysis based on clustering combined with the label of "age" (Fig. 2a). Most Sertoli cells in 2-year-olds were identified as Stage_a cells, and the ratio between the numbers of Sertoli cells in Stage_a and Stage_b declined with age and reached the lowest point at puberty (11 years). Stage_c Sertoli cells did not appear until after the age of 11 years and were predominant in

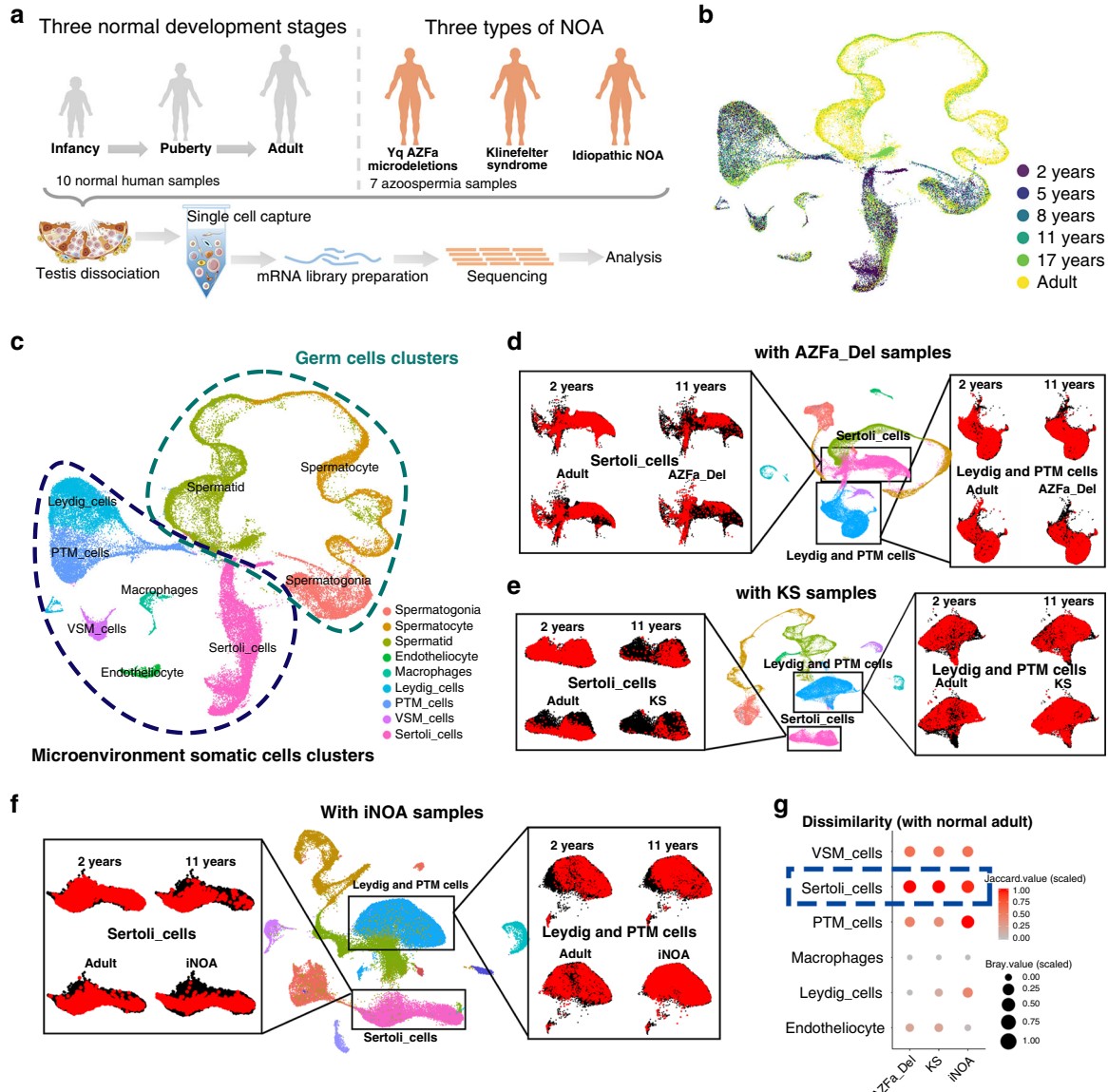

**Fig. 1 Global expression profiling of human testicular cells from infancy to adulthood and in NOA patients by single-cell RNA-seq. a** Schematic illustration of the experimental workflow. **b, c** UMAP plots of all testicular cells from 10 healthy subjects. Cells are colored for **b** ages or **c** types. UMAP, uniform manifold approximation and projection. **d–f** UMAP plots of all testicular cells from 10 healthy subjects merged with **d** 1 case of AZFa_Del, **e** 3 cases of KS, or **f** 3 cases of iNOA samples. Sertoli cells (left panels) or Leydig&PTM cells (right panels) are isolated and highlighted as red according to the sample type. **g** Dissimilarity of somatic cells between normal adult and AZFa_Del, KS or iNOA are shown on bubble diagram. The gradient of bubbles sizes indicates low to high scaled Bary value, and the gradient of red indicates low to high scaled Jaccard values. Some elements in panel **a** were downloaded from Servier Medical Art repository. Note: peritubular myoid cells (PTM_cells), vascular smooth muscle cells (VSM_cells); Yq AZFa microdeletions (AZFa_Del); Klinefelter Syndrome (KS); idiopathic NOA (iNOA).

the late puberty and adult. It is worth noting that the full range of Sertoli cell stages was observed in adult testes (Fig. 2c), suggesting that the stepwise maturation of Sertoli cells starts at a very young age, and reaches a steady state in late puberty. To balance the potential sampling bias, we reconciled our sequence data with two previous studies[9,11]. The pseudotime analysis of the merged data also showed clear three stages pattern (Supplementary Fig. 3a, b) and similar development trend (Supplementary Fig. 3c).

Next, we further analyzed the dynamic changes in the gene expression pattern at each stage. The total number of expressed genes and total UMI count per cell declined dramatically from Stage_a to Stage_b and Stage_c (Supplementary Fig. 3d). 989, 561, and 1086 DEGs were observed in three states, respectively

(Supplementary Data 2 and Fig. 2d). *EGR3*, *JUN*, and *NR4A1* were the top three DEGs in Stage_a, *S100A13*, *ENO1*, and *BEX1* were the top three DEGs in Stage_b, and *HOPX*, *DEFB119*, and *CST9L* were the top three DEGs in Stage_c (Fig. 2e). To verify this result, we performed IHC staining of testis sections from subjects of different ages for JUN, ENO1, and DEFB119 (Fig. 2f, g). Then, we performed GO analysis of DEGs at each stage. In Stage_a cells, the main enriched GO terms were "stem cell differentiation," "cell fate commitment," and "maintenance of cell number" (Supplementary Fig. 3e), suggesting that Sertoli cells at this stage exhibit some characteristics of stem or progenitor cells. In Stage_b, "small molecule catabolic process", "generation of precursor metabolites and energy" and "cellular amino acid metabolic process" were enriched (Supplementary

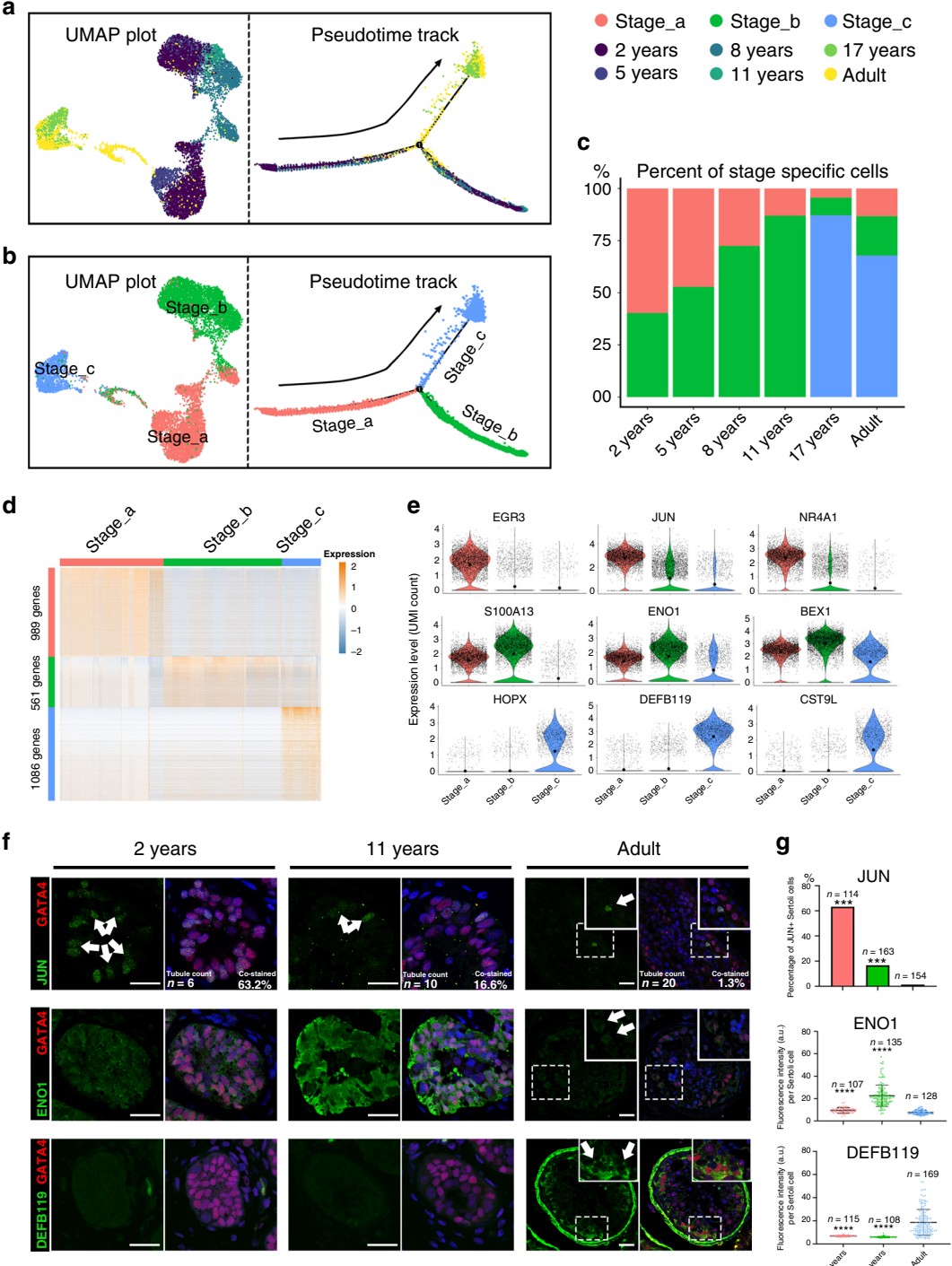

Fig. 3f). In Stage_c, the main enriched GO terms were "cellular metabolic compound salvage", "protein transmembrane transport" and "phagosome maturation" (Supplementary Fig. 3g), indicating that the main functions of mature Sertoli cells are phagocytosis of germ cells and their metabolite.

Then, we proceeded to annotate basic physiological characteristics, and compared the proliferation and the energy metabolism type of the three stages. Based on the previously reported S and G2/M phase-specific genes[14], we found Sertoli cells in Stage_a expressed higher levels of mitotic genes than other two latter stages (Supplementary Fig. 3h, i). As regards heterogeneity in energy metabolism, there was no obviously difference of three types of metabolism-related genes (glycolysis, oxidative

phosphorylation, and triglyceride metabolism) between Sertoli cells and other somatic cells. When focus on Sertoli cells, all three types of metabolism-related genes downregulated from Stage_a to Stage_c (Supplementary Fig. 3j), however, when considering the percentage of gene expression, the glycolysis-related genes showed an opposite trend (Supplementary Fig. 3k). It indicated that immature Sertoli cells were in an active metabolic state which dominated by oxidative phosphorylation, and the mature Sertoli cells were in a low level of energy metabolism mainly through glycolytic pathways.

All these changes in expression profile, proliferation, and energy metabolism revealed that Sertoli cells went through three distinct consecutive developmental stages.

**Fig. 2 Identification of three maturation stages during Sertoli cell development. a, b** Analysis of Sertoli cells (UMAP plot and pseudotime trajectory plot), with cells colored by **a** age or **b** stage. **c** Bar plot showing the proportion of Sertoli cells at each stage (Stage_a, red; Stage_b, green; Stage_c, blue) in each age group. **d** Heatmap showing the DEGs of each stage during Sertoli cell maturation. DEG counts are shown on the left of the color bar of the cell type annotation. **e** Violin plot showing the expression levels of the top DEGs at each stage (Stage_a, red column; Stage_b, green column; Stage_c, blue column). The larger black dots in each violin plot column represent the mean value. **f** Immunofluorescence co-staining of GATA4 (red) with JUN (green, upper panel), ENO1 (green, middle panel), and DEFB119 (green, lower panel) in human testicular paraffin sections at three ages. The scale bar represents 20 μm. ***$p < 0.001$. (comparing with normal adult). **g** The statistics of the percentage of JUN+ Sertoli cells (JUN+ /GATA4+ co-staining cells), and the fluorescence intensity of ENO1 and DEFB119 per Sertoli cell (GATA4+ cells). The percentage of JUN+ Sertoli cells in adult testis was significantly lower than that in 2 years ($p = 0.0002$) and 11 years ($p = 0.0003$); the fluorescence intensity of ENO1 per Sertoli cell in adult was significantly lower than that in 2 years ($p = 2.3E-06$) and 11 years ($p = 4.7E-08$); the fluorescence intensity of DEFB119 per Sertoli cell in adult testis was significantly higher than that in 2 years ($p = 1.6E-09$) and 11 years ($p = 6.2E-08$). In the upper panel (JUN), data shown as barplot of the percentage of JUN+ Sertoli cells in three age based on 5 fields, statistical analysis made by chi-square test; in the middle and bottom panel (ENO1 and DEFB119), data shown as mean ± SD, statistical analysis between fluorescence intensity per Sertoli cell in adult testis and immature testis made by two-tailed, unpaired non-parametric test with Mann–Whitney test; the confidence interval is 95%. ***$p < 0.001$, ****$p < 0.0001$ (comparing with normal adult).

**Regulatory networks during sertoli cell maturation**. We hypothesized that the levels of key regulators should change dramatically at the junction between two consecutive stages. Therefore, we screened 372 candidate regulators that showed a significant change at a branch point of the pseudotime trajectory (Fig. 3a). GO terms associated with these genes included "TGF-beta signal pathway", "tube development", and "response to steroid hormone" (Fig. 3b), indicating steroid hormones are one of the main upstream extracellular regulatory signals. The heatmap of "response to steroid hormone"-related genes is shown in Fig. 3c. Then, we identified pathways that potentially regulate the maturation of Sertoli cells by ingenuity pathway analysis (IPA). IPA analysis showed that from Stage_a to Stage_b, cell proliferation-related signaling such as "ERK5 signaling", "IGF-1 signaling" and "EGF signaling", etc. were inhibited, while "remodeling of epithelial adherens junctions", etc. were activated. In addition, "Unfolded protein response" enriched in the early two stage, it may be caused by a higher expression level of proteins and the lack of organelles, such as the endoplasmic reticulum in immature Sertoli cells (Fig. 3d). From Stage_b to Stage_c, "cardiac hypertrophy signaling" and "Wnt/β-catenin signaling", etc. were inhibited and "germ cell–Sertoli cell junction signaling", etc. were activated (Fig. 3e). Furthermore, Rho relative signaling pathways ("RhoA Signaling", "Regulation of Actin-based Motility by Rho" and "RhoGDI signaling") which were reported to promote reorganization of the actin cytoskeleton and regulate cell shape, attachment, and motility were also activated in the two later stages. In the latter two stage, VDR/RXR/PPAR relative signaling pathways ("VDR/RXR Activation" and "PPAR Signaling" in Stage_b, and "PPARα/RXRα Activation" and "Superpathway of Cholesterol Biosynthesis" in Stage_c), which regulate the cholesterol and lipid metabolism, were enriched. These results indicate that the proliferation of Sertoli cells mainly occurs in Stage_a, and structural remodeling start at Stage_b.

To explore the key regulators and identify the transcriptional regulatory network during Sertoli cell maturation, we analyzed 1,665 human transcription factors using the Algorithm for the Reconstruction of Accurate Cellular Networks (ARACNe)[15], and the ssmarina package in R was used to analyze their downstream gene set. We found that FOSB, EGR3, KLF10, etc. might specifically play a critical role in Stage_a Sertoli cells (Fig. 3f, g), whereas HMGB1, SUB1, CNBP, MEF2C, ZFP36L2, NFIX, etc. might be the master regulators in Stage_b Sertoli cells (Fig. 3f–i). In Sertoli cells in Stage_c, RORA, SMARCA1, and HOPX were the top candidate gene expression regulators (Fig. 3h, i). These results were confirmed by staining testis sections for EGR3 and HOPX (Fig. 3j, k). These results provide insight in potential regulatory networks, ranging from the upstream hormonal signals to intracellular regulatory pathways.

**The multi-lineage interactome network and its dynamic changes in the spermatogenic microenvironment**. To investigate the complex signaling networks and their dynamic changes in the spermatogenic microenvironment, we performed an unbiased ligand–receptor interaction analysis between these testicular cell subsets by CellphoneDB[16]. In Stage_a, Stage_b, and Stage_c, we found 158, 96, and 72 interactions between Sertoli cell ligands and receptors from other cells and 189, 49, and 41 interactions between ligands from other cells and Sertoli cell receptors, respectively (Supplementary Fig. 4a–c). The interaction between Sertoli cells and other testicular cells showed unique changes in different stage and cell type (Supplementary Fig. 4d). Focusing on the interaction between Sertoli cells and spermatogonia, we found 12/99 interaction pairs involved in TGF-β signaling, was the highest proportion. The expression of the ligands of TGF-β signaling decreased rapidly after puberty (Supplementary Fig. 4e, f, i). In addition, TGF-β signaling pathways may regulate both germ and somatic cells in a cross-network manner, for example, INHA was produced by Sertoli cells, while its receptors, ACVR2B and TGFBR3 were expressed in spermatogonia and Leydig cells, respectively (Supplementary Fig. 4e, f). Except for TGF-β signaling, some important pathways for spermatogonial self-renewal in rodent, such as GDNF-GFRa1/Ret and FGFs-FGFRs signaling also showed an expression pattern that target to spermatogonia (Supplementary Fig. 4g, h). Interestingly, in most interaction pairs between germ cells and Sertoli cells, Sertoli cells play the role of signal source, whereas in most interaction pairs between Leydig&PTM cells and Sertoli cells, Sertoli cells receive signals in the early stage. IGF and NOTCH signaling between Sertoli cells and Leydig&PTM cells play important roles in the early stage of spermatogenic microenvironment maturation (Supplementary Fig. 4d). Interactions between collagen (COL1A2) in Sertoli cells and integrins α1β1, α10β1, and α2β1 in endotheliocytes were detected at the early stage. In addition, Sertoli cells also expressed the ligands of CD74, which is mainly expressed in macrophages; this interaction occurs mainly in Stage_b (Supplementary Fig. 4d).

These findings indicate complex paracrine regulatory networks of the maturation of the human spermatogenic microenvironment.

**Single-cell profiling revealed that different types of NOA testes have distinct features of Sertoli cell defects**. An altered microenvironment has been reported in NOA patients in previous studies[4,17], but the underlying pathological mechanisms remain unknown. Therefore, we re-clustered 10 normal samples with samples from 3 iNOA patients, 3 KS patients, and 1 AZFa_Del patient to investigate the changes of the microenvironment under pathologic states (Supplementary Fig. 5a, b). Interestingly, we

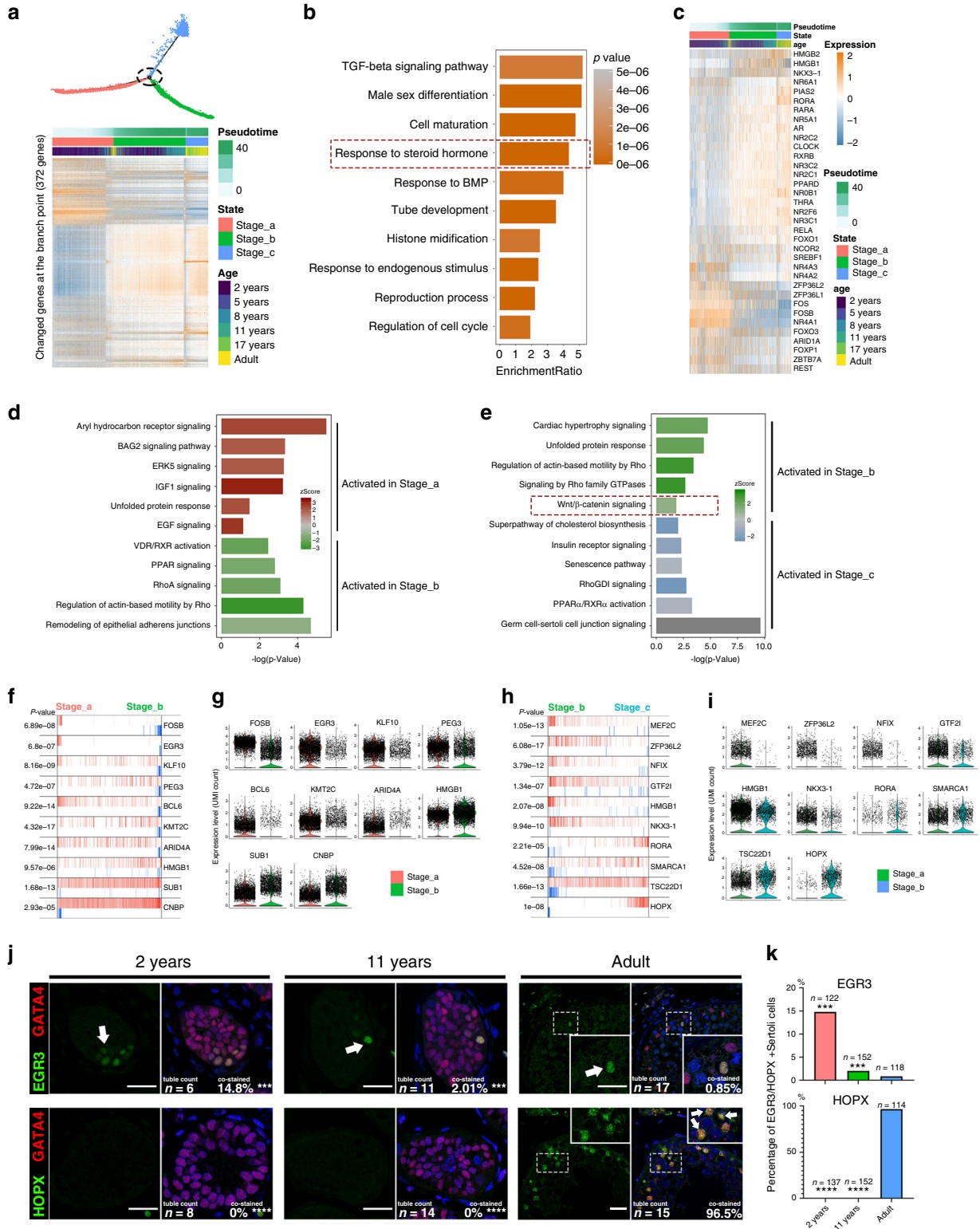

found that samples from different pathological types showed significant differences, whereas low heterogeneity was observed among samples belonging to the same type (Supplementary Fig. 5c). In addition, the quality of single-cell sequencing libraries (total UMI count and number of expressed genes per cell) in all three pathological types was within an acceptable range (Supplementary Fig. 5d).

Focusing on Sertoli cells, we found that part of the iNOA Sertoli cells overlapped with normal immature Sertoli cells;

however, Sertoli cells of AZFa_Del and KS patients were close to but did not overlap with any normal immature samples in the UMAP reduced-dimension plot (Fig. 4a, b). Pseudotime trajectory analysis showed that iNOA Sertoli cells were considerably heterogeneous, as one cluster was in Stage_a and another in Stage_b; AZFa_Del Sertoli cells were in the early part of Stage_c; and KS Sertoli cells were arrested in Stage_b (Fig. 4b). The expression level of cell cycle specific genes in iNOA Sertoli cells were higher than those of other adult subjects (Fig. 4c), suggesting

**Fig. 3 Inferred signal pathways and master regulators for three stages of Sertoli cells. a** Heatmap of 372 genes that exhibited dramatic changes in gene expression at the intersection of two consecutive stages of Sertoli cell development. **b** Pathways and biological process terms of GSEA are shown as barplot. The GSEA score is presented on the x-axis, and the gradient of orange indicates low to high FDR values. **c** Heatmap of 37 genes related to the GO term "response to steroid hormone." **d, e** Pathway terms of IPA enriched at each stage are shown as barplot. The P-value is presented on the x-axis, and the gradient of two colors indicates low to high activation scores in the two stages. **f–h** The top 10 candidate master regulators at each stage of Sertoli cell development identified by the master regulator analysis algorithm (MARINa) and **g, i** violin plots of the expression levels of those regulators in two consecutive stages. In the MARINa plots, activated targets are colored red and repressed targets are colored blue for each potential master regulator (vertical lines on the x-axis). On the x-axis, genes were rank-sorted by their differential expression in two consecutive developmental stages. The p values on the left indicate the significance of enrichment, calculated by permutating two developmental phases. **j** Immunofluorescence co-staining of GATA4 (red) with a master regulator of Stage_a, EGR3 (green, upper panel), and a master regulator of Stage_c, HOPX (green, lower panel), in human testicular paraffin sections at three ages. The scale bar represents 20 μm. (comparing with normal adult). **k** The statistics of the percentage of EGR3+ (upper panel) or HOPX+ (lower panel) Sertoli cells. The percentage of EGR3+ Sertoli cells in adult testis was significantly lower than that in 2 years ($p = 0.0002$) and 11 years ($p = 0.0004$), and the percentage of HOPX+ Sertoli cells in adult testis was significantly lower than that in 2 years ($p = 3.53E-07$) and 11 years ($p = 7.11E-06$). Data shown as barplot of the percentage of EGR3+ or HOPX+ Sertoli cells in three age based on 5 fields, statistical analysis made by chi-square test (according to the number of staining positive and negative cells); the confidence interval is 95%. ***$p < 0.001$, ****$p < 0.0001$ (comparing with normal adult).

a higher proliferation rate in iNOA Sertoli cells. In addition, Sertoli cells in the three types of NOA showed energy metabolism patterns that were different from that of healthy adults. The expression levels of glycolysis-related genes in both KS and AZFa_Del Sertoli cells were lower than that in iNOA but higher than that in normal adult. The expression levels of oxidative phosphorylation-related genes in all three types of NOA Sertoli cells were higher than that in normal adult. Triglyceride metabolism is important for Sertoli cells to perform physiological functions. Our data showed that iNOA and AZFa_Del Sertoli cells have a higher triglyceride metabolism level than KS and normal adult (Fig. 4d). Some important regulators of Sertoli cell development were also shown to be differently expressed between NOA patients and healthy subjects; for example, expression of the Stage_a regulator gene *FOSB* was higher in iNOA and AZFa_Del, and *EGR3* was highly expressed in iNOA, while the transcriptional activity of the Stage_c regulators *HOPX* and *TSC22D1* in iNOA was lower than in other adult samples (Fig. 4e). This result was corroborated by immunofluorescence co-staining of HOPX with the Sertoli cell marker GATA4; HOPX expression levels in iNOA Sertoli cells were lower than that in other patients and healthy adults (Fig. 4f, g).

**Sertoli cells of KS and AZFa_Del patients exhibited abnormal gene expression patterns**. KS and AZFa_Del patients have clear etiologies (disorder of the sex chromosomes at the gene or chromosome level), but unknown pathogenesis for spermatogenic dysfunction. Therefore, we analyzed the differences in sex chromosome gene expression patterns of these two NOA types and compared them with those of healthy subjects to analyze potential mechanisms of pathogenesis. PAS staining showed that the AZFa_Del sample was Sertoli cell-only syndrome (SCOS) with a Johnsen score of 2 (all germ cells in seminiferous tubules lost with only Sertoli cells left) (Supplementary Fig. 1d). In total, 1286 and 1426 DEGs were down- and upregulated in AZFa_Del Sertoli cells, respectively, compared with control adult Sertoli cells (Supplementary Data 3 and Supplementary Fig. 6a). Gene set enrichment analysis (GSEA) terms such as "chromosome localization" and "double-strand break repair" were decreased in AZFa_Del, while "humoral immune response" and "regulation of immune system process" were enriched (Supplementary Fig. 6b). In addition, genes such as macrophage migration inhibitory factor (*MIF*) and defensin beta 119 (*DEFB119*), involved in cell-mediated immunity, immunoregulation, and inflammation, were also highly expressed in AZFa_Del (Supplementary Fig. 6c), suggesting aberrant immune activation is involved in the pathogenesis of AZFa_Del Sertoli cells. Interestingly, the expression

percentage of X chromosomal but not Y chromosomal genes in AZFa_Del Sertoli cells was different from healthy adults (Supplementary Fig. 6d). The AZFa region contains three genes, i.e., *USP9Y*, *DBY*, and *UTY*, among them, *USP9Y* was highly expressed in Sertoli cells (Supplementary Fig. 6e). All three AZFa genes were highly expressed before puberty and were down-regulated with age. It is worth noting that *UTY* transcript was detected in AZFa_Del Sertoli cells, *UTY* is located in chromosome Y (13230770-13480670) and not covered by the two AZFa test sites (sY84 and sY86), so this patient was an incomplete AZFa region microdeletion with *UTY* gene existing (Supplementary Fig. 6f).

In KS testes, most seminiferous tubules suffered more severe atrophy than AZFa_Del, with Johnsen scores of 0–1 (all germ cells in seminiferous tubules lost and the Sertoli cells were also damaged or even lost), and exhibited vascular basement membrane thickening combined with severe interstitial proliferation, while the remaining seminiferous tubules contained few Sertoli cells with abnormal morphology, as shown by testicular biopsy (Supplementary Fig. 1c), indicating the abnormal X chromosomes affected both germ cells and Sertoli cells. Total 831 and 1130 DEGs were down- and upregulated in KS Sertoli cells compared with healthy adults (Supplementary Data 4 and Supplementary Fig. 6g). Interestingly, immune response-related terms such as "response to interferon-alpha" and "humoral immune response" were positively enriched in KS, similar to AZFa_Del (Supplementary Fig. 6h). In addition, *MIF* and beta-2-microglobulin (*B2M*) were found among the top DEGs in KS (Supplementary Fig. 6i), indicating aberrant immune activation might be a universal pathogenic process resulting from sex chromosome dysfunction. A subset of X-linked genes escapes silencing by X-inactivation and is expressed from both X chromosomes. Therefore, we analyzed escaping X-linked genes (eXi genes) and non-escaping X-linked genes (neXi genes) separately[18]. The expression of eXi genes in KS was higher than in healthy adults in all eight types of testicular cells (Supplementary Fig. 6j). Higher expression levels of neXi genes were observed in germ cells and Sertoli cells of KS patients, but these differences were not found in endotheliocytes, macrophages, and Leydig&PTM cells (Supplementary Fig. 6j). The fold changes of eXi genes and neXi genes were calculated (Supplementary Fig. 6l). With respect to Sertoli cells, the expression of both eXi genes and neXi genes in KS Sertoli cells was higher than in healthy adults (Supplementary Fig. 6k).

**Sertoli cell profiling revealed signals involved in maturation defects in iNOA patients.** The etiology of iNOA could not be

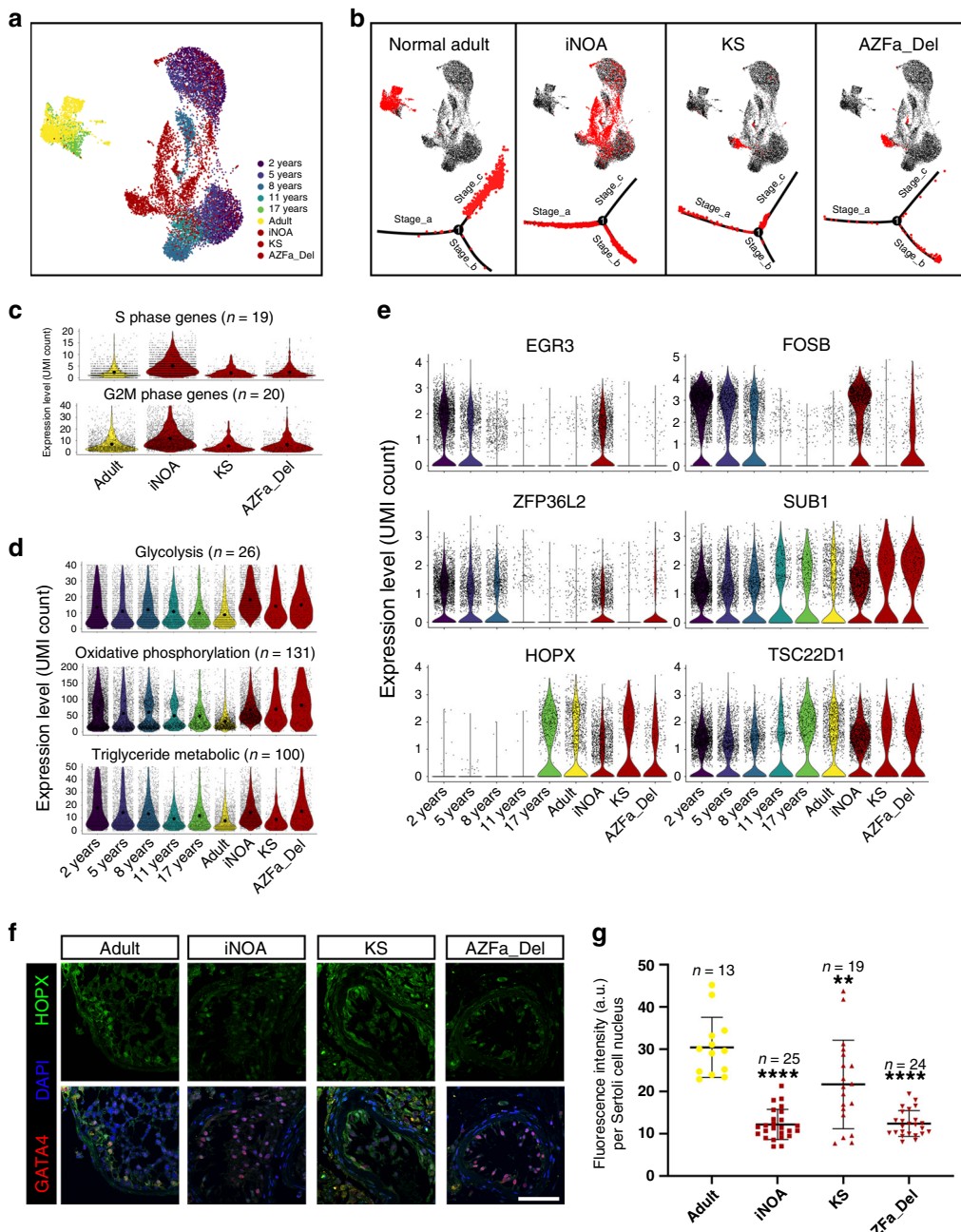

**Fig. 4 Single-cell transcriptome profiling of three types of NOA Sertoli cells. a** Analysis of normal Sertoli cells at six ages merged with iNOA, AZFa_Del, and KS Sertoli cells, in a UMAP plot with cells colored by age and sample type. **b** Healthy adult, iNOA, AZFa_Del, and KS Sertoli cells are highlighted red in the UMAP plot (upper row) and the pseudotime trajectory plot (lower row). **c** Violin plot of the expression levels of S phase and G2M phase-specific genes in healthy adults and three types of pathological Sertoli cells. The larger black dots in each violin plot column represent the mean value. **d** Violin plot of the expression levels of energy metabolism-related genes in healthy (six different ages) and three types of NOA Sertoli cells. The larger black dots in each violin plot column represent the mean value. **e** Violin plot of the expression levels of stage-specific master regulators in healthy (six different ages) and three types of NOA Sertoli cells. **f, g** Immunofluorescence co-staining of GATA4 (red) and HOPX (green) in normal adult and three types of pathological testicular paraffin sections (**f**). The statistics of nucleoplasm located HOPX expression is shown as histogram (**g**). The fluorescence intensity per Sertoli cell in adult testis was significantly higher than iNOA ($p = 7.32E-05$), KS ($p = 0.0077$) and AZFa_Del ($p = 8.21E-05$).The scale bar represents 50 μm. Data shown as mean ± SD, statistical analysis between fluorescence intensity per Sertoli cell in normal adult and three types of NOA testis made by two-tailed, unpaired non-parametric test with Mann–Whitney test; the confidence interval is 95%. **$p < 0.01$, ****$p < 0.0001$ (comparing with normal adult).

explained by genetic abnormalities of sex chromosomes or otherwise. Therefore, we further investigated the iNOA Sertoli cells, and focusing on the heterogeneity of maturation, the proliferation ability, and their regulatory network. PAS staining of testicular biopsies of all 3 iNOA patients revealed typical SCOS with Johnsen scores of 2; vascular basement membrane

thickening and a higher proportion of interstitial cells were also observed, but these were not as severe as in KS (Supplementary Fig. 1c). The UMAP plot of Sertoli cells from 5 healthy adults and 3 iNOA patients showed that these Sertoli cells could be further clustered into three subsets, i.e., clusters 1 and 2 (composed of iNOA Sertoli cells) and cluster 3 (healthy adult Sertoli cells)

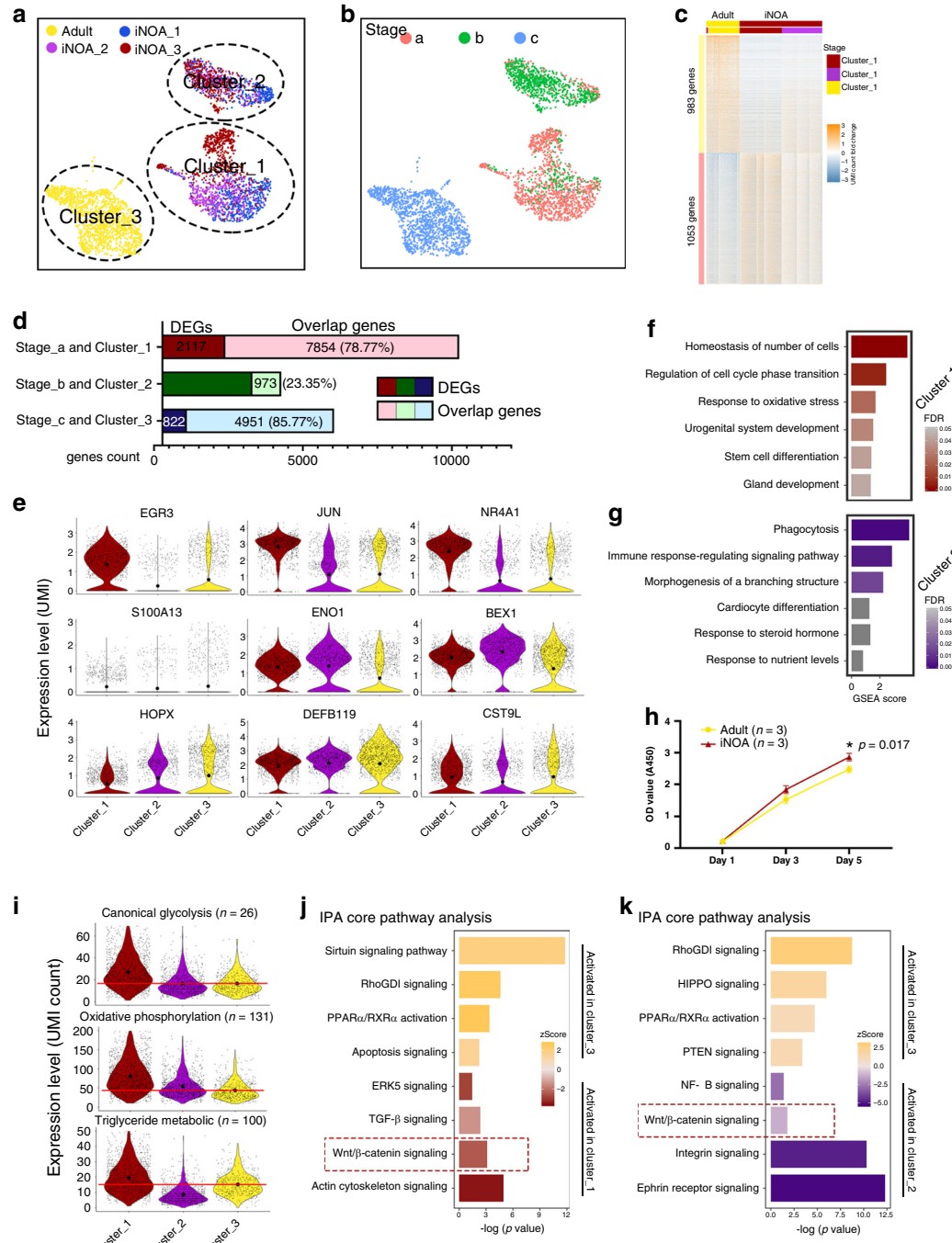

**Fig. 5 The changes in expression pattern showed the heterogeneity and maturation arrest of iNOA Sertoli cells. a**, **b** UMAP plot showing the heterogeneity among iNOA Sertoli cells; a clear difference is observed between iNOA and normal adult Sertoli cells. **c** Heatmap of DEGs between healthy adult and iNOA Sertoli cells. **d** The percentage of overlap genes between iNOA (cluster 1 and cluster 2) and normal developmental Sertoli cells in Stage_a or Stage_b, and between normal adult (cluster 3) and other normal Sertoli cells in Stage_c. **e** Violin plot of the expression levels of candidate stage-specific markers in three Sertoli cell clusters. The upper row shows Stage_a markers; the middle row shows Stage_b markers, and the bottom row shows Stage_c markers. The larger black dots in each violin plot column represent the mean value. **f**, **g** GSEA terms enriched in **f** cluster 1 and **g** cluster 2 iNOA Sertoli cells, shown as barplot. The GSEA score is presented on the x-axis, and the gradient of red indicates low to high FDR values. **h** iNOA Sertoli cells proliferated faster than normal adult Sertoli cells in vitro, as shown by CCK-8 assay. Data shown as mean ± SD from three biological independent samples and two times independent experiments, statistical analysis made by two-tailed, Student's t-test; the confidence interval is 95%. *p < 0.05 (comparing with normal adult). **i** Violin plot of the expression levels of energy metabolism-related genes in three Sertoli cell clusters. The larger black dots in each violin plot column represent the mean value. **j**, **k** Pathways terms of IPA enriched in **j** cluster 1 and **k** cluster 2 compared with normal adult Sertoli cells, shown as barplot. The p-value is presented on the x-axis, and the gradient of two colors indicates low to high activation scores in the two clusters.

(Fig. 5a). Pseudotime trajectory analysis showed the maturation characteristics of cluster 1 and cluster 2 were close to those of Stage_a and Stage_b Sertoli cells, respectively (Fig. 5b). The heterogeneity of iNOA Sertoli cells was confirmed by the heatmap of DEGs between healthy adults and iNOA patients (Supplementary Data 5 and Fig. 5c). When comparing Sertoli cells in Stage a, b, and c with that in cluster 1, 2, and 3, respectively, we found Stage_a and cluster 1 shared 7854 overlap genes, which account for the 78.77% of their total expressed genes. However, only 973 (23.35%) overlap genes in Stage_b and cluster 2 was identified (Fig. 5d), indicating that cluster 1 contained immature Sertoli cells that resembled Stage_a cells (normal immature Sertoli cells), but cluster 2 contained immature Sertoli cells with a pathological transcription profile. The top DEGs of each stage during Sertoli cell development coincided, at least partially, with the cluster-specific expression patterns of the three clusters (Fig. 5e). Next, we performed GSEA using the DEGs of clusters 1 and 2. The GSEA terms which were enriched in normal Stage_a Sertoli cells, such as "maintenance of cell number" and "stem cell differentiation," were also enriched in cluster 1 iNOA Sertoli cells (Fig. 5f), indicating the immaturity of cluster 1 Sertoli cells. The enriched GSEA terms in cluster 2, such as "phagocytosis," "immune response-regulating signaling pathway," and "response to steroid hormone," also indicated cluster 2 shows similarity with Stage_b cells at some level (Fig. 5g).

We found that iNOA Sertoli cells were more proliferative (Figs. 4c and 5h). Regarding the energy metabolism pattern, energy metabolism characteristics of clusters 1, 2, and 3 were similar to those of Stage_a, Stage_b, and Stage_c Sertoli cells, respectively (Fig. 5i). All these results indicate a maturation arrest in part of the iNOA Sertoli cells.

To better understand the mechanisms underlying maturation arrest and aberrant transcription, we compared cluster 3 with cluster 1 and cluster 2, and then performed core IPA with the DEGs. Pathway terms such as "ERK5 signaling," "TGF-β signaling," and "Wnt/β-catenin signaling," were enriched in cluster 1 (Fig. 5j). In addition, pathway terms such as "RhoGDI signaling" and "PPARα/RXRα activation" were decreased in clusters 1 and 2, suggesting the silencing of these two pathways may mediate a pathological transformation in iNOA Sertoli cells. We found that "integrin signaling," "ephrin receptor signaling," and "Wnt/β-catenin signaling" were activated in cluster 2 iNOA Sertoli cells (Fig. 5k). In addition, we also identified the activation of Wnt/β-catenin signaling with the β-catenin staining in Sertoli cell nucleus (Supplementary Fig. 7a, b). Considering Wnt/β-catenin signaling was inhibited in healthy Stage_b and Stage_c cells (Fig. 3d, e) and iNOA Sertoli cells showed reduced maturation, we hypothesized that the aberrant activation of the Wnt pathway may cause maturation defects in iNOA Sertoli cells.

**Inhibition of the Wnt pathway restored the maturation defects of Sertoli cells in iNOA patients**. We found mature Sertoli cells would de-differentiate and "re-enter" into Stage_a when isolated from testicular tissue and cultured in vitro. These changes manifested as follows: (1) mature Sertoli cells were in the resting phase but regained proliferation ability in vitro (Fig. 5h); (2) the expression of stage-specific markers, such as HOPX and JUN, more closely resembled that of Stage_a Sertoli cells (Figs. 2f and 6a); and (3) iNOA Sertoli cells were arrested in a relatively naive state in vivo, but they showed no difference in maturation compared with normal adult cells when cultured in vitro. Importantly, we noticed Wnt/β-catenin signaling was activated in immature and iNOA Sertoli cells (Figs. 3e and 5j–k) and was inhibited in normal mature Sertoli cells, suggesting the Wnt pathway may play an important role in the regulation of Sertoli

cell maturation. To investigate this hypothesis, we cultured iNOA and normal adult Sertoli cells in vitro and treated them with three kinds of Wnt signaling pathway inhibitor, ICG-001 (ICG), XAV939, and KY02111[19–21] (Fig. 6b), we found ICG showed the highest efficiency to inhibit the nucleus translocation of β-Catenin (Supplementary Fig. 8a) and Sertoli cell proliferation (Supplementary Fig. 8b). In addition, the classical Wnt target genes such as CYCLIND1, JUN, and CD44 were downregulated by ICG treatment, so ICG was chosen for further experiment (Supplementary Fig. 8c). After ICG treatment for 14 days, the ICG-treated Sertoli cells showed a larger cell surface, reduced refraction, and an altered internal texture (Fig. 6c), but also kept a high percentage (87.5–92.1%) of SOX9 staining and a low percentage (0–4%) of smooth muscle actin (SMA) staining (Supplementary Fig. 8d). In addition, CCK-8 results showed that ICG treatment strongly decreased the proliferation of both iNOA and normal adult Sertoli cells (Fig. 6d).

We next assessed the effects of ICG treatment on the mRNA levels of stage-specific genes by quantitative real-time PCR (qPCR). We found that Stage_a-specific genes, such as FOSB, EGR3, and JUN, were downregulated after ICG treatment, while the mRNA levels of Stage_c-specific genes, such as HOPX and TSC22D1, were significantly higher than in control cells (Fig. 6e). However, the expression of DEFB119, a highly DEG in Stage_c, was not changed after ICG treatment (Fig. 6e), indicating that this gene may be regulated by other stimuli during Sertoli cell development. We further checked the expression changes at the protein level. IHC staining results showed that the proportion of HOPX-positive cells in the ICG-treated group was significantly higher than that in the DMSO treated group (Fig. 6f, g). However, the proportion of JUN shown an opposite trend and did not show any co-staining with HOPX (Fig. 6f, g). In addition, the expression levels of INHA and AMH, two ligands of the TGF-β signaling pathway, were low in Stage_c cells, as evidenced by IHC staining and ELISA (Fig. 6h, i).

To determine whether the maturity of Sertoli cells affected their support function for germ cells, we used iNOA and normal adult Sertoli cells with or without IGC treatment as feeder cells, and then seeded GS cells, which is a mouse germ cell line mainly composed of SSCs and type A spermatogonia[22]. From the 5th to 9th day, we observed more and larger GS cell colonies in the ICG-treated group than in the control group (Fig. 6j and Supplementary Fig. 8e, f). There was no difference in cell colony size between GS cells fed by ICG-treated iNOA cells and ICG-treated normal adult Sertoli cells, but in the GS cells fed by non-treated Sertoli cells, cell colonies fed by normal adult Sertoli cells were larger than in the iNOA group, but the difference was not significant (Supplementary Fig. 8e, f). All GS cells were collected and counted on the 9th day. The counts of viable GS cells in the ICG-treated groups ($2.97 \times 10^5$ in the normal adult Sertoli group and $3.21 \times 10^5$ in the iNOA Sertoli group) were also significantly higher than DMSO treated groups ($1.02 \times 10^5$ in the normal adult Sertoli group and $0.53 \times 10^5$ in the iNOA Sertoli group) (Fig. 6k), in addition, the expression of SSCs markers did not change in the ICG treated group (Supplementary Fig. 8g), indicating inhibition of the Wnt pathway could alleviate the maturation disorder of iNOA Sertoli cells and improve the supporting function of both iNOA and normal Sertoli cells in vitro (Fig. 7).

**Discussion**
The role of the testicular somatic microenvironment for germ cells is like the role of soil for sprouting seeds, providing them with various necessary nutrients. Although an increasing body of evidence suggests that the human testicular cells are highly heterogeneous, our understanding of testicular homeostasis and

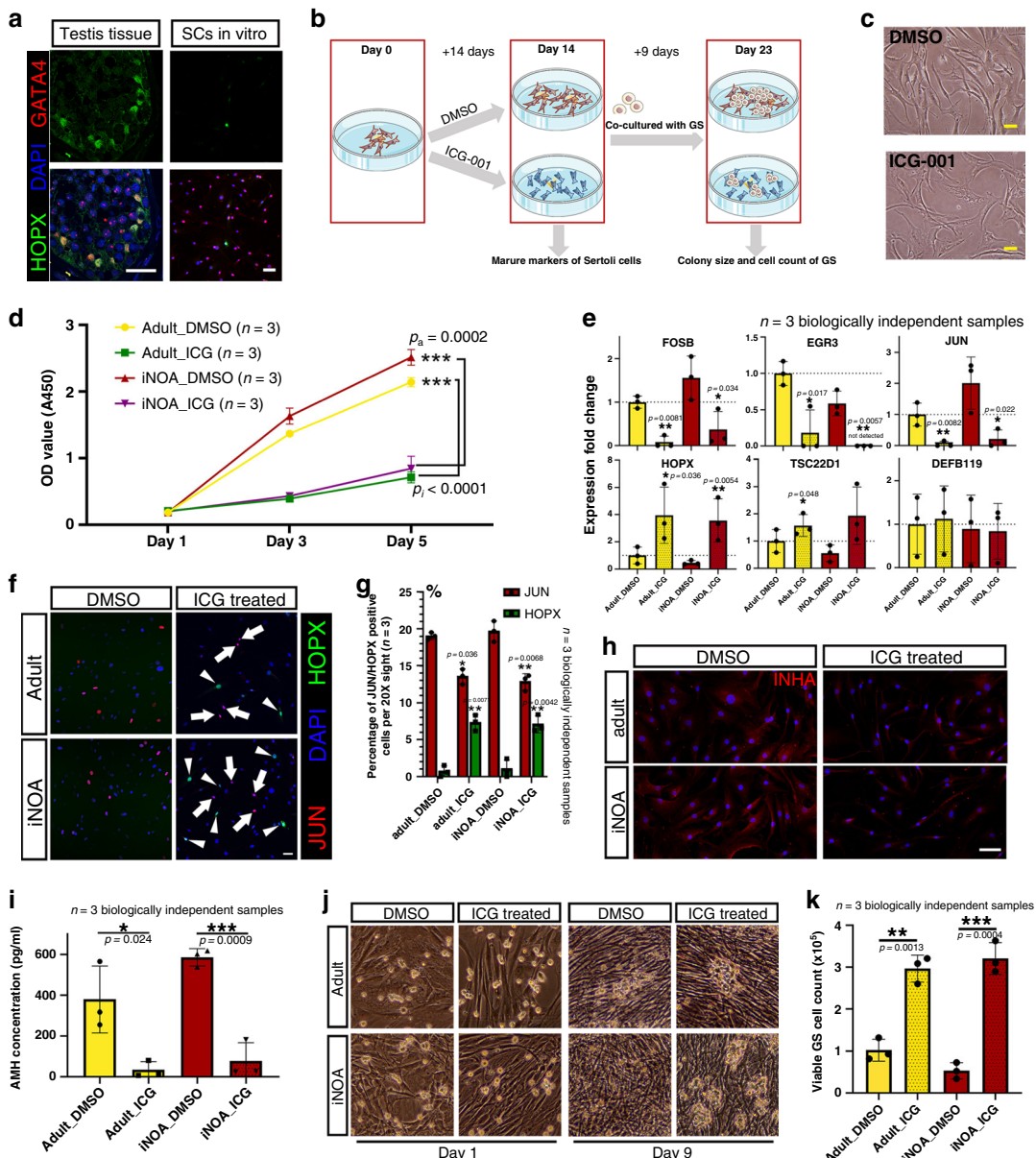

**Fig. 6 Inhibition of the Wnt pathway induced the maturation of Sertoli cells in vitro. a** Immunofluorescence co-staining of GATA4 (red) and HOPX (green) in normal and pathological testicular paraffin sections and cultured Sertoli cells. The scale bar represents 40 μm. **b** Schematic illustration of the Wnt pathway inhibition experiment. **c** The morphology of cultured Sertoli cells with or without ICG treatment. The scale bar represents 20 μm. **d** Sertoli cells treated with ICG proliferated faster than DMSO-treated controls in vitro, as indicated by CCK-8 assay. Data shown as mean ± SD from three biologically independent samples and two times independent experiments, statistical analysis made by two-tailed, Student's t-test; the confidence interval is 95%. ***p < 0.001 (comparing with normal adult). **e** qPCR results showing the expression fold change of six stage-specific markers in cultured iNOA and normal adult Sertoli cells with or without ICG treatment. The gene expression levels of normal adult Sertoli cells without ICG treatment (DMSO treated) were used as the baseline values. Data shown as mean ± SD from three biological independent samples. Statistical analysis made by two-tailed, unpaired Student's t-test; the confidence interval is 95%. *p < 0.05, **p < 0.01, ***p < 0.001 (comparing with normal adult). **f, g** Immunofluorescence co-staining of JUN (red) and HOPX (green) in cultured iNOA and normal adult Sertoli cells with or without ICG treatment (**f**). The scale bar represents 20 μm. The statistics of JUN/HOPX positive cells count is shown as histogram (**g**). Data shown as mean ± SD from three 20X sights. This experiment was repeated two times independently with similar results. Statistical analysis made by two-tailed, unpaired Student's t-test; the confidence interval is 95%. *p < 0.05, **p < 0.01, ***p < 0.001 (comparing with DMSO treated group). **h** Immunofluorescence of INHA (red) in cultured iNOA and normal adult Sertoli cells with or without ICG treatment. The scale bar represents 20 μm. **i** ELISA results showing the difference in AMH concentration in the supernatant of cultured iNOA and normal adult Sertoli cells with or without ICG treatment. Data shown as mean ± SD from three independent samples. Statistical analysis between ICG and DMSO treatment group made by two-tailed, unpaired Student's t-test; the confidence interval is 95%. *p < 0.05; ***p < 0.001 (comparing with DMSO treatment group). **j, k** The morphology of cultured GS cell colonies when normal adult or iNOA Sertoli cells with or without ICG treatment are used as feeder cells (**j**). The statistics of viable GS cells count is shown as histogram (**k**). Images were taken with a ×20 magnification microscope. Data shown as mean ± SD from three independent samples. Statistical analysis between ICG and DMSO treatment group made by two-tailed, unpaired Student's t-test; the confidence interval is 95%. **p < 0.01; ***p < 0.001 (comparing with DMSO treatment group).

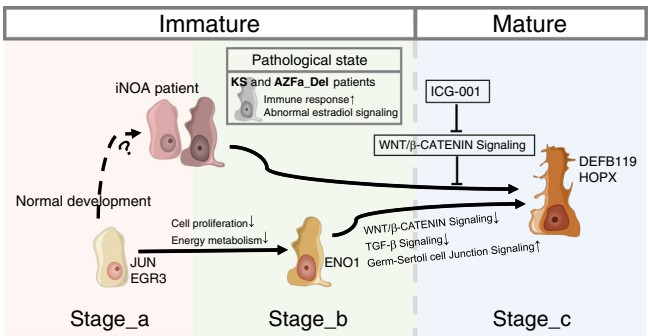

**Fig. 7 The schematic diagram of the main content of this study.** The schematic diagram of this study showed the normal developmental changes of Sertoli cells and their alterations in the three types of NOA patients. The images of Sertoli cells were downloaded from Servier Medical Art repository.

disorders remains hampered by our understanding of cell type complexity and cell-to-cell interactions, especially for somatic cell development. No effective treatments exist for NOA, especially iNOA. This is probably because most NOA patients are not diagnosed until childbearing age, when spermatogenesis has already been irreversibly damaged, and no or few germ cells are left. However, the other testicular cell types, with distinct physiological and pathological characteristics at the molecular level, are also completely absent. In the present study, we compared the developing testes from NOA patients with those of healthy subjects at an unparalleled resolution. Our study is a big step towards the understanding of the testicular microenvironment and male infertility-related diseases.

Most human testicular cell populations functionally develop after birth[11]. However, little is known about the development of human testicular cell lineages from newborns to puberty. Our first important finding is that we shed light on the cellular processes underlying the development of the testicular microenvironment by taking developmental time as the horizontal axis. Sertoli cells serve as the scaffold in the microenvironment. Previous studies suggested that Sertoli cells go through only two developmental stages, i.e., from immature to mature[6,23]. Sertoli cells remain immature until the peak of testosterone production during puberty[23,24]. A recent study base on single-cell sequencing data of pre- and post- puberty testis reported that there are two immature Sertoli cells states that converge into a single mature state[9]. However, in this study, we found that the cell count ratio between Stage_a and Stage_b was decreased with age, but Stage_a cells were more proliferative than Stage_b. Therefore, we thought Sertoli cells more likely go through three independent and consecutive developmental stages (Stage_a to Stage_b to Stage_c pattern). In infancy and early childhood, Sertoli cells of Stage_a are predominant and proliferate vigorously, with characteristics similar to those of stem cells. This may be due to various factors secreted by early Leydig cells, such as IGF[25]. Rapid amplification helps to increase the volume of seminiferous tubules. Before puberty, proliferation of many Sertoli cells is significantly reduced, and they enter Stage_b. At this stage, Sertoli cells express many genes to prepare for structural maturation. From a metabolic perspective, we observed that once a cell enters Stage_b, its preferred metabolic pathway changes from oxidative phosphorylation to glycolysis. This is consistent with the previous observation that lactic acid (produced by glycolysis) provides energy for germ cell development[26]. It is worth noting that all three stages of Sertoli cells exist and form a steady state in adult testes, maintaining the basic structure of seminiferous tubules and also guaranteeing homeostasis of the microenvironment for

spermatogenesis for decades. Having defined the populations in the testicular microenvironment, we also revealed an unbiased ligand–receptor interaction network among different cell types, providing a more holistic overview. Focusing on Sertoli cells, we investigated the processes underlying the maturation of the microenvironment by analyzing single-cell transcriptomes from humans of different ages.

We systematically compared the construction of the microenvironment in healthy subjects with that in NOA patients. We found that the most common defects observed in NOA patients are dysfunctional Sertoli cells. Other important findings include the main features of Sertoli cell defects in patients with different types of NOA. Both KS and AZFa_Del are caused by a sex chromosome disorder. Because of X-inactivation[18], we observed that the total levels of X chromosomal genes in each type of testicular cell in KS are less than twice that in healthy adults. On the other hand, we found Sertoli cells express the highest proportion of X-inactivated genes, indicating Sertoli cells are more susceptible to dysregulation due to an extra X chromosome than other testicular cells. In AZFa_Del patients, although the loss of *DDX3Y* has been reported to cause dysfunction in germ cells, we found *USP9Y*, another AZFa region gene, was specifically expressed in Sertoli cells, indicating the pathological change of AZFa_Del Sertoli cells was not only caused by the loss of germ cells, but also by AZF region deletion. In addition, both KS and AZFa_Del Sertoli cells abnormally expressed high levels of *MIF*, a gene encoding a cytokine that stimulates the immune response and induces apoptosis in immune as well as non-immune cells[27]. These findings indicate that in NOA patients with genetic disorders in sex chromosomes, Sertoli cells proceed to the mature stage, but a chronic immune response may induce cell death.

In patients with iNOA, we found that Sertoli cells were observably immature and heterogeneous. In addition, over half of the DEGs in cluster 1 overlapped with Stage_a, suggesting that these cells physiologically closely resemble healthy immature Sertoli cells. In cluster 2, we observed an upregulation of genes involved in (i) the immune response pathway and (ii) some Stage_b-specific features, such as structural morphogenesis and the response to steroid hormone. It should be noted that the germ cells themselves also affect the cells in the microenvironment. Could the immaturity of Sertoli cells in iNOA patients simply be caused by the loss of germ cells? We think the answer is negative, for the following two reasons. First, Sertoli cells in AZFa_Del patients without germ cells could nevertheless enter Stage_c. Second, in co-culture experiments with germ cells, we are not able to induce the maturation of patients' Sertoli cells. Restoring the maturity of the microenvironment may activate local spermatogenic foci. This prompted us to understand the maturation process of Sertoli cells and to screen for diagnostic biomarkers and regulators that could potentially serve as therapeutic targets for iNOA treatment.

Surprisingly, Sertoli cell transcriptomes of patients with iNOA are highly similar, although the possible causes are unknown. Even if the initial causative factors are unclear, finding pathways to improve the microenvironment may still be helpful for disease treatment. We identified candidate pathways and regulators which may play a crucial role during Sertoli cell maturation. The activating signals are mainly stem cell maintenance-related signals, such as the Wnt/β-catenin, ephrin receptor, and integrin pathways. This is consistent with the immature phenotype of Sertoli cells. While the Wnt/β-catenin pathway was activated in all patients, we also found that a negative regulator of the Wnt/β-catenin pathway, HOPX, was not expressed in Stage_a Sertoli cells but highly expressed in Stage_c cells. HOPX is considered to promote stem cell maturation by inhibiting Wnt signaling[28]. Furthermore, a rodent study showed that constitutive Wnt/β-

Catenin signaling in Sertoli cells leads to their continuous proliferation and compromised differentiation, resulting in increased germ cell apoptosis and infertility[29]. By treating cells with an inhibitor that suppresses Wnt signaling, we successfully promoted the partial maturation of some Sertoli cells derived from patients, and these cells could even maintain the proliferation of spermatogonial stem cells. This is also the first step for the treatment of such patients. Until this proves successful, NOA treatment methods remain very limited, and effective treatment targets are still lacking.

However, RNA-seq cannot answer all the details of the entire process. Some functional maturation cannot be evaluated by transcriptome analysis. Orthogonal validation approaches, such as IHC in testes of different ages and gene overexpression/ knockdown experiments in animal models, are required. Considering most changes happen in puberty, more intensive sampling around puberty may provide more detailed information about the developmental processes and transitions. In addition, there is another possibility that germ cell loss at different developmental stages in the different NOA cases results in distinct responses from the Sertoli cells, for example, earlier loss causing maturation defects and later loss resulting in inflammation. Even though, this hypothesis contradicted the results of GS cells coculture, using human spermatogonia in future study to exclude the species difference would add credibility to the conclusion. Likewise, future studies with larger sample sizes of KS, AZF_Del, and iNOA patients at different ages, especially around puberty, could strongly improve our understanding of the causes underlying spermatogenic dysfunction in these diseases.

In summary, our study not only revealed the major cell types and the processes underlying their maturation processes in normal human testes, but also compared the maturation disorders of testicular cells in the main types of NOA patients. These comparisons shed light on the pathogenesis of different types of NOA and pave the way for the development of diagnostic and treatment methods for NOA.

## Methods

**Contact information for reagent and resource sharing**. Requests for further information, resources, and reagents should be directed to and will be fulfilled by the Lead Contact, Zheng Li (lizhengboshi@sjtu.edu.cn) or the first author, Liangyu Zhao (zlytsg@sjtu.edu.cn).

**Experimental model and subject details**. The experiments performed in this study were approved by the Ethics Committee of Shanghai General Hospital (License No. 2016KY196). For human testis samples, all participants (and their legal guardian if aged <18 years) signed their consent after being fully informed of the goal and characteristics of our study. Fresh testicular tissues were obtained from 5 male donors who underwent testicular biopsy or partial excision for the following indications: contralateral testis to testicular torsion ($n = 1$), benign testicle mass ($n = 3$), or contralateral testis to cryptorchidism ($n = 1$), and an additional 5 OA and 7 NOA samples were obtained from the abandoned tissues after testicular sperm extraction operation. Ten normal samples (5 underage and 5 OA donors) all had normal karyotypes, genotypes, sex hormone levels, and morphology of seminiferous tubules according to their age. The KS and AZFa microdeletion donors were diagnosed by spectral karyotyping and qPCR. The qPCR examination before hospitalization showed that in this sample, sY84 and sY86 were completely deleted. Among all donors, other abnormal genotypes related to spermatogenic disorders were excluded by whole-exome sequencing.

**Histological examination**. Fresh testicular tissues from donors were fixed in 4% paraformaldehyde for 12–24 h at 4 °C, embedded in paraffin, and sectioned. Before staining, tissue sections were dewaxed in xylene, rehydrated using a gradient series of ethanol solutions, and washed in distilled water. Then the sections were stained with PAS/hematoxylin and dehydrated using increasing concentrations of ethanol and xylene. Sections were allowed to dry before applying neutral resin to the coverslips. The staining images were captured with a Nikon Eclipse Ti-S fluorescence microscope (Nikon). Johnsen score of testis sections was identified according to the previous study[30].

**Immunohistochemical staining and fluorescence intensity measure**. The sections were rehydrated by the same method as for PAS/hematoxylin staining. After rehydration, sections were processed for antigen retrieval with 10% sodium citrate at 105 °C for 10 min. Tissues were blocked with 5% normal donkey serum and incubated with appropriate primary antibodies (Supplementary Data 6) at 4 °C overnight. Sections were further incubated with secondary antibody for 2 h at room temperature. Nuclei were labeled with DAPI by incubating tissue sections for 15 min. Images were captured with an OLYMPUS IX83 confocal microscope.

The measure of fluorescence intensity was processed with Photoshop CS6. For the fluorescence intensity of JUN, ENO1, HOPX, and β-catenin in testis sections, we first chose the cell regions with GATA4 positive staining or their nuclear regions, and then converted the single staining of target protein to grayscale. After that, the fluorescence intensity of chosen regions were recorded. For the Wnt/β-catenin activation in cultured Sertoli cells, we first chose the cell nucleus regions according to DAPI staining, then the average fluorescence intensity of β-catenin staining in each chosen regions of DMSO treated group were calculated as the threshold. Then, the inhibitors treated groups were measured with the same method. Cells whose nuclear fluorescence intensity were higher than the threshold were thought to be activated.

**Isolation of single testicular cells**. Testicular cells were isolated from human and mouse testicular tissues by enzymatic digestion according to a previously described method. In brief, testicular tissues were enzymatically digested with 4 mg/ml collagenase type IV, 2.5 mg/ml hyaluronidase, and 1 mg/ml trypsin at 37 °C for 20 min. Given the risk of damaging certain types of big cells (such as Sertoli cells), cut-up of the tissue was avoided during this process. Subsequently, the cell suspension was filtered through a 40-mm nylon mesh, and the cells were sorted by MACS with a Dead Cell Removal Kit (Miltenyi Biotec) to remove dead cells. The cells were resuspended in 0.05% BSA/PBS buffer before 10× Genomics library preparation.

**Isolation and culturing of human Sertoli cells from testicular tissue**. Isolation and identification of human Sertoli cells were according to our previous study[31,32]. Briefly, Testicular tissues were washed three times and then cut the testicular tissues into 0.2 cm × 0.2 cm pieces and incubated with Enzyme I (10 mL of DMEM containing 2 mg/mL type IV collagenase and 10 mg/mL DNase I) at 34 °C for 15 min. After that, seminiferous tubules were washed again and incubated with Enzyme II (4 mg/mL collagenase IV, 2.5 mg/mL hyaluronidase, 2 mg/mL trypsin, and 10 mg/mL DNase I) at 34 °C for 10–15 min. Tissue block were filtered by a 40-mm cell strainer and the mix of germ cell and Sertoli cells were cultured in DMEM/F-12 supplemented with 10% FBS at 34 °C in 5% $CO_2$. After 24 h, the suspending germ cells were removed and Sertoli cells were attached to culture dishes. More details should be obtained in the Protocol Exchange DOI: 10.21203/rs.3.pex-997/v1.

**Single-cell RNA-seq library preparation**. Cell suspensions were loaded on a Chromium Single Cell Controller instrument (10× Genomics, Pleasanton, CA, USA) to generate single-cell gel beads in emulsions (GEMs). Single-cell RNA-seq libraries were prepared using the Chromium Single Cell 3′ Library & Gel Bead Kit (P/N 120237, 10× Genomics) according to the manufacturer's instructions. Briefly, suspensions containing about 8000 cells per sample were mixed with RT-PCR reaction before being added to a chromium chip already loaded with barcoded beads and partitioning oil. The chromium chip was then placed in a Chromium Single Cell Controller instrument. GEM-RT-PCR was performed in a C1000 Touch Thermal cycler with 96-Deep Well Reaction Module (Bio-Rad; CT022510) to produce barcode cDNA using the following program: 53 °C for 45 min; 85 °C for 5 min; maintain at 4 °C. Barcoded cDNA was isolated from the partitioning oil and then amplified by PCR. Sequencing libraries were generated from amplified cDNA using a 10× chromium kit, including reagents for fragmentation, sequencing adaptor ligation, and sample index PCR. Final libraries were sequenced on an Illumina Novaseq 6000.

**Mapping, sample quality control, and integration**. Cell Ranger software (version 2.2.0), provided by 10× Genomics, was used to demultiplex cellular barcodes, map reads to the genome and transcriptome using the STAR aligner, and downsample reads as required to generate normalized aggregate data across samples, producing a matrix of gene counts versus cells. After mapping, the filtered count matrices of each sample were tagged with a special library batch ID, and a *Seurat* object was created using the Seurat package in R. Cells were further filtered according to the following threshold parameters: the total number of expressed genes, 500–9000; total UMI count, between $-\infty$ and 35,000; and proportion of mitochondrial genes expressed, <40%. Normalization was performed according to the package manual (https://satijalab.org/seurat/v3.1/pbmc3k_tutorial.html). Because cell capturing with LZ013/LZ014/LZ015 and LZ017/LZ018/LZ019 was performed with a BD Rhapsody system with three samples in one batch, these six samples could be regarded as two independent batches. Other samples captured with 10× Genomics were considered as 11 independent batches. Batch correction was performed using the IntegrateData function in the Seurat package.

**Cell identification and clustering analysis**. The merged Seurat objects were scaled and analyzed by principal component analysis (PCA). Then the first 20 principal components (PCs) were used to construct a KNN graph and refine the edge weights between any two cells. Based on all these cells' local neighborhoods, the FindClusters function with the resolution parameter set as 0.1 was used to cluster the cells. In total 16 clusters were identified, and these clusters were renamed by accepted marker genes. The first 20 PCs were also used to perform non-linear dimensional reduction by UMAP, and the dimension reduction plots were given as output (Fig. 1c). For further analysis, we isolated Sertoli cells and performed these two steps again; we obtained three clusters.

**Cluster-cluster dissimilarity analysis**. By following Seurat pipeline, 3 germ cell clusters and 6 somatic clusters were obtained based on which we merged normal and NOA samples and did tech/batch correction. We grouped these cells into sub-clusters by both cell type (9 clusters) and sample type (normal adult, AZFa_Del, KS and iNOA). We then calculated the average expression of the top 1000 DEGs (they were also used to generate the UMAP plot in Fig. 1) in each sub-cluster and scaled these DEGs to balance the difference of total expression level among sub-clusters. After that, we calculated the Jaccard and Bray-curtis distance between each two sub-clusters base on the scaled values with Vegan 2.5.6 R package, and the Jaccard and Bray-curtis distance between same cell type of two sample type were showed as bubble chart.

**Identification of differentially expressed genes**. The Seurat FindAllMarkers function (test.use = wilcox) is based on the normalized UMI count to identify unique cluster-specific marker genes. Unless otherwise noted, only the genes that were detected in at least 10% of the cells were tested, and the average $\log_2$(fold change) threshold was set as 1 in the analysis of eight testicular cell types and as 0.25 in the subgroup analysis of Sertoli cells. Unless otherwise noted, the DEGs in each selected sub-cluster were calculated based on comparison between this sub-cluster and the rest of the dataset, and the GESA analysis were based on the DEGs which come from comparation between the two corresponding groups. GO analysis was performed using WebGestalt 2019 (http://www.webgestalt.org/#).

**Cell trajectory analysis**. Single-cell pseudotime trajectories were constructed with the Monocle 2 package (version 2.8.0) according to the operation manual (http://cole-trapnell-lab.github.io/monocle-release/docs_mobile/). Briefly, the UMI count matrices of the Sertoli cells and Leydig&PTM cells were input as the expr_matrix, and meta.data was input as the sample sheet. Then, 11,521 ordering genes (DEGs among clusters) were chosen to define a cell's progress. In this step, ordering genes that were expressed in less than 10 cells and had a P-value bigger than 0.001 were excluded; batch effects were removed by setting the residualmodelFormulaStr parameter. DDRTree was used to reduce the space down to one with two dimensions, and all cells were ordered with the orderCells function.

**Cellular energy metabolism analysis**. Genes associated with glycolysis, oxidative phosphorylation, or triglyceride metabolism were obtained from the AmiGO 2 database (http://amigo.geneontology.org/). These genes were input into Seurat, and the proportions of the genes were calculated by the PercentageFeatureSet function.

**Transcription factor network construction**. A total of 1469 human transcription factors in AnimalTFDB and a gene expression matrix were taken as input for the ARACNe-AP software (version 1.0.0). The results of ARACNe-AP were input into the ssmarina (version 1.01) R package, which further calculated the marina objects containing the normalized enrichment scores, P-values, and a specific set of regulators.

**Analysis of inactivated X chromosomal genes in the AZFa-Del sample**. A list of inactivated X chromosomal genes was obtained from Carrel's study[18]. Active and inactive X chromosomal genes were input into the Seurat object, and their proportions were calculated by the PercentageFeatureSet function.

**Cell proliferation assays**. Cell proliferation was analyzed by the expression level of cell cycle phase-specific gene or CCK-8 assay. The cycle phase-specific genes list were obtained from previous study[14], genes which expressed in less than 1% cell were excluded. For CCK-8 assay, 3000 Sertoli cells were seeded in a 96-well microplate containing 200 μl culture medium per well. After cell attachment, the cells were starved in serum-free medium for 16 h. Then, the medium was replaced with 10% FBS DF12 medium with 10 μM ICG-001 (ICG treatment) or DMSO (control). For comparison between adult and iNOA Sertoli cells, the medium was replaced with 10% FBS DF12 medium. After 1–5 days of culturing, CCK-8 medium (Dojindo) was added to the cells and incubated for 3 h. The optical density (OD) for each well was measured at 450 nm using a microplate reader (Bio-Rad Model 550).

**GS cell supporting assays**. Sertoli cells from OA or iNOA patients were first treated with or without 10 μM ICG-001 in DF12 medium for 14 days. Then $10^5$

Sertoli cells from each group were re-seeded in 6-well microplates as the feeder cells. After 24 h, $10^5$ GS cells were seeded in each well, and the medium was replaced with GS culture medium[22]. The growth of cells was recorded daily under ×20 magnification field. After 9 days, GS cells were digested by 0.25% pancreatin for 2 min, and the GS cells were gently aspirated. Digested GS cells were cultured for another 1 h to remove potential Sertoli cells, and stained with Trypan blue before counting. The change of GS cells markers were identified by q-PCR. The software used for collecting qPCR data was Bio-Rad CFX manager 3.1.

**AMH secretory ability assay**. The AMH concentration in Sertoli cell supernatant was measured with the Human Anti-Mullerian Hormone (AMH) ELISA Kit (JYMBIO) according to the manufacturer's instructions.

**Statistics and reproducibility**. Unless otherwise noted, all data are represented as mean ± standard deviation (SD). Statistical significance was determined in GraphPad Prism 8 software by using non-parametric test (for the result in Fig. 4g and Supplementary Figs. 7b, 8b, f; unpaired; Mann–Whitney test; two-tailed), Student's t-test (for results in Fig. 6e, g and Supplementary Fig. 8c, g; unpaired; two-tailed) or chi-square test (for result in the upper panel of Fig. 2g and two panel of Fig. 3j and Supplementary Fig. 8a) and the confidence interval is 95%. Results were considered to be significant at $p < 0.05(*)$. Statistical parameters are reported in the respective figures and figure legends. The experiments results in Figs. 2f, 3j, 4f, 6a, c, f, h, j, k and Supplementary Figs. 4i, 7a were repeated two times independently with similar results. In addition, testicular tissue samples from children and AZFa_Del patient are difficult to obtain, so the PAS and IHC staining for each age and NOA type (Figs. 2f, 3j, 4f and Supplementary Figs. 1d, e, 4i, 7a) in this study was based on only one biologically repeat, but the CCK8, qPCR, ELISA, ICC result which used OA and iNOA adult samples were based on 3 or 4 biological independent samples.

**Reporting summary**. Further information on research design is available in the Nature Research Reporting Summary linked to this article.

## Data availability
The RNA-seq matrix data generated in this study has been deposited in NCBI with accession number GSE149512 (this study). Other sequencing datasets used can be found in NCBI GEO under accession numbers GSE134144 and GSE124263 (for Supplementary Fig. 3a–c). The list of TFs can be obtained in Supplementary Data 6 of this study or from http://bioinfo.life.hust.edu.cn/AnimalTFDB/#!/. The annotation of gene function analysis can also be obtained in Supplementary Data 6 of this study, from http://amigo.geneontology.org/amigo/landing/, or from GitHub (https://github.com/zlyingithub/17-testis-single-cell-R-github-data). All other relevant data supporting the key findings of this study are available within the article and its Supplementary Information files or from the corresponding authors upon reasonable request. Source data are provided with this paper.

## Code availability
All code associated with this manuscript and the gene annotation list have been uploaded to GitHub (https://github.com/zlyingithub/17-testis-single-cell-R-github-data).

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

## Acknowledgements

We are grateful to Dr. Lingsong Li and Dr. Yumiko Saga for reading the paper. This work was support by grants from National Key R&D Program of China (2017YFC1002003 and 2018YFA0107702); National Natural Science Foundation of China (81671512, 81701524, 81871215, and 81701428); the ShanghaiTech University start-up fund of Zhi Zhou; Doctorial innovation fund of Shanghai Jiao Tong University School of Medicine (BXJ201940); Project funded by China Postdoctoral Science Foundation (2019M661524). We wish to thank the (1) Shanghai OE Biotech CO. LTD, (2) Sinotech Genomics CO. LTD. Shanghai and (3) Novogene Bioinformatics Institute, Beijing, China for their support of RNA-seq library preparation. We thank Dr. Jianming Zeng and his bioinformatics team for their reference codes.

## Author contributions

Z.L., Z.Z., C.W., and J.S. conceived the project. L.Z., C.Y., X.X., T.J., C.Y., J.Z., J.L., N.L., Z.D., X.L., J.F., and N.L., performed the experiment. P.L., Z.Z., R.T., and H.C. collected the samples. L.Z. conducted the bioinformatics analyses. L.Z., C.Y., and T.J. wrote the paper with help from all of the authors.

## Competing interests

The authors declare no competing interests.
