## [Peer Review File · Nature Communications]

Reviewers' Comments:

Reviewer #1:

Remarks to the Author:

Zhao LY., et al used single-cell RNA-seq to characterize the transcriptome of 10 healthy donors and 7 Non-obstructive azoospermia (NOA) patients. They found that idiopathic Sertoli cells ("nurse" cell of the testicles that helps in the process of spermatogenesis) in NOA (iNOA) patients are immature, and this characteristic is unique to iNOA. Interestingly, inhibiting Wnt pathways can promote maturation of Sertoli cells, suggesting a potential treatment for iNOA. This is an interesting manuscript and the authors did very solid work for validating their findings.

Major comments:

Figure 2 A (left): Based on the UMAP plot, the distance between adult cells (yellow color) and stage_a (2-5 years) is shorter than the distance between Stage_a and stage_b.

But in Figure 4A, the adult cells are far away from cells of 2-5 years. Any explanation for this?

The three stages are the fundamental finding in the manuscript. The pattern is very clear but hope can have more validations to confirm that these patterns are not due to data sampling bias (e.g., gaps in ages, population differences (e.g., family, geography differences)).

Overall it is a nice work.

Minor comments:

The manuscript is overall well written but some expressions need to be changed.

E.g.,

"We have a superficial understanding that Sertoli cell development after birth is mainly divided into two stages: the immature and mature stage."

A better way to express this can be something like "Although most studies divided Sertoli cell development after birth into two stages: the immature and mature, it is largely unknown whether there exists any intermediate or transition states" -- a more positive way to acknowledge current limitations.

Reviewer #2:

Remarks to the Author:

This manuscript by Zhao et al., describes a single cell (sc)RNA-Seq analysis of human testis cells from healthy juveniles and adults plus a set of individuals suffering germ cell defects and infertility (NOA). Initial analysis by the authors suggested pronounced defects in Sertoli cell (SC) function in NOA adults and the manuscript is primarily focused on characterising SC development in healthy individuals and SC defects in NOA patients. SCs are known to be essential for germ cell maintenance and fertility. Mining of scRNA-Seq data is supported by testis immunostaining and SC culture experiments. By analysis of SC gene expression during testis development, a developmental pathway is suggested of immature to mature stages. Data from NOA patients with sex chromosome defects indicated that SC maturation was intact but increased expression of inflammatory genes may underlie germ cell loss. The idiopathic (i)NOA cases showed defects in SC maturation associated with potentially increased canonical Wnt pathway activation. Culture-based experiments provide evidence to support a role for Wnt signalling in SC maturation (healthy and iNOA samples) and targeting this pathway with a small molecule (ICG) enhanced features of SC maturation. Treatment with ICG also appeared to improve the ability of cultured SCs to support mouse germline/spermatogonial stem cell growth. The analysis therefore highlighted the Wnt pathway as potential therapeutic target for iNOA individuals, assuming early intervention prior to germ cell loss.

Mechanisms underlying germ cell loss and infertility in males, even with specific genetic lesions,

are poorly understood. This manuscript therefore addresses an important and under-studied area and can be an important resource for the reproduction field. The focus on SCs is appropriate given their fundamental role in supporting spermatogenesis and the identified link between specific defects in SC function and infertility is of great interest. However, while the datasets are very valuable and presented well, subsequent analysis is in places superficial or lacks quantification and requires further study and explanation. Similarly, the culture-based analysis (Fig. 6) is of great interest but preliminary and needs additional characterization.

Points to be addressed are below:

1. While the manuscript focus is on SCs and other somatic cells, the scRNA-Seq identifies germ cell populations from spermatogonia to spermatids. The authors perform some immunostaining to highlight stem (GFRA1) and differentiating spermatogonia (KIT) in testis samples from different ages (Fig S1). Multiple recent papers have delineated different spermatogonial subtypes in human testis by scRNA-Seq (e.g. from the groups of Bradley Cairns and Miles Wilkinson). Importantly, these studies find that the most primitive spermatogonia express low levels of GFRA1 and rather are marked by high levels of TSPAN33, UTF1, PIWIL4 etc. Given the many single cell studies now published, it would be informative if the authors can correlate their spermatogonia data to these other studies and check expression of other stem cell markers by immunostaining. Text should be modified to include these points.
2. From UMAP analysis of healthy and NOA samples in Figure 1, it was concluded that "Sertoli cells showed greater difference in spatial distribution than other populations." Eyeballing the figure does suggest this but this is a subjective statement. Can this be supported by quantitative analysis?
3. From pseudotime analysis of healthy SCs (Figure 2), the authors conclude that these cells go through a defined developmental pathway (stage a to b to c) with increasing age. Confusingly, the pseudotime plot shows a bifurcation from a to c and a to b stages. Why is this? Interestingly, a recent study of human testis during puberty by the Cairns group (Guo et al. Cell Stem Cell 2020, 26:262) suggests that there are two immature SC states that converge into a single mature state. The authors need to reconcile their datasets with this previous study and clarify the developmental pathway supported by the data.
4. In Figure 2G the authors immunostain testis sections with a range of identified markers of the distinct SC stages to support the proposed developmental pathway. This dataset needs to be quantified and significance of the GATA4 counterstain mentioned in the figure legend. The immunostained proteins have different localisation patterns – are these as expected? Some further details on these marker proteins should be provided.
5. From GO analysis of DEGs of different SC stages (Figure S3), the authors highlight a selection of terms including "stem cell differentiation" and cell fate commitment" for stage "a" SCs. Other GO terms were also found but not mentioned e.g. "rhythmic processes" and "covalent chromatin modification". It seems that the terms that suited the proposed developmental model were focused on but other potentially important terms were ignored. This section should be discussed in more detail.
6. Mitotic phase prediction of different SC stages (Figure S3) was performed and stage "a" cells were concluded to be most proliferative. However, no significance value is provided with this analysis. Are the differences significant?
7. IPA analysis of SCs suggests potential pathways important at different developmental stages (Figure 3). The authors highlight a limited number of pathways in the text and refer to the others as "et al". Other identified pathways have seemingly been ignored/de-emphasized but the rationale for this is unclear. This same point applies to identification of transcription factors important for each stage on the next page. The text should be revised to better describe the

available data. What is the relevance of "cardiac hypertrophy signaling" for stage "b" cells and why does this indicate that structural remodeling mainly occurs at that point?

8. For immunostaining data of Figure 3K, how many tubule cross-sections were scored? This information needs to be included in the figure legend and/or text.

9. From analysis of ligand-receptor interaction between SCs and spermatogonia, "most" pairs involved TGF- β signalling (Figure S4). The authors should indicate in the text the percentage of pairs. Was GDNF-GFR α 1/Ret signalling identified? Are any FGFs identified? Rodent studies indicate key roles for both pathways in spermatogonial self-renewal. On a related note, Figure S4F is not described in the text. What is the relevance of these panels?

10. Cell cycle prediction of SCs from iNOA patients indicated enhanced proliferation (Figure 4C). Is this change in proliferation vs healthy individuals significant? Confusingly, while percentage of iNOA SCs predicted to be in S-phase was increased, those in G2/M were decreased and those in G0/G1 similar to controls. Can this also suggest that iNOA cells are arrested in S-phase? If cells are cycling more it might be predicted that both S and G2/M phases are increased while G1/G0 cells decreased. This point should be clarified. On a related note, most metabolic pathways are predicted to be downregulated in iNOA SCs but these cells are concluded to be more active/proliferative. Given the energy demands of cell cycle, this conclusion seems counter-intuitive.

11. From HOPX immunostaining in Figure 4F it is concluded that iNOA SCs show reduced expression of this maturation marker but quantification is not provided. The percentage of SCs positive for this marker should be included.

12. At multiple points, the authors refer to the "Johnsen" score of testis pathology. For the benefit of general readers, the meaning of the scores should be highlighted in the text e.g. no germ cells present.

13. The authors state that low expression of UTY was detected in AZFa-del SCs suggesting that multiple copies of this gene are present outside this deleted Y chromosome region. Presumably this must be known? Can the authors elaborate on this point?

14. Detailed analysis of iNOA SCs (Figure 5) suggests different maturation states which are then compared to stages a,b and c from analysis in Figure 4B. The different iNOA SC stages are concluded to "resemble" stages a and b. However, in Figure 4B these cells have already been plotted onto stage a and b regions of the pseudotime trajectory so presumably are identifiable as these stages. Can the authors clarify the relationship between analysis in Figure 4 and 5?

15. From a culture-based assay of iNOA SCs, it is concluded that these cells are more proliferative than healthy SCs (Figure 5H). However, the difference in growth rate, although statistically significant, appears very mild. Would this change in proliferation rate be biologically significant? On a related note, further details of how this in vitro assay was performed should be included in the text.

16. A role of the Wnt pathway in SC maturation is concluded from analysis of scRNA-seq data and treatment of cultured SCs with Wnt pathway inhibitor ICG (Figure 6). Is the ICG specific to the Wnt pathway or are there expected off-target pathways? ICG is reported to bind the common coactivator CBP and some effects on cells may be independent of effects on the Wnt pathway (e.g. Arensman et al., PMID: 25082960). Are classical Wnt target genes affected in SCs by treatment with ICG? What does NC refer to in the figures? On a related note, studies in rodents have demonstrated that constitutive Wnt/ β -catenin signalling in SCs blocks their maturation (Tanwar et al. *Biology of Reproduction* 2010, 82: 422-432) – in line with the authors conclusions. However, this study is not mentioned. This study should be highlighted in the text and compared to the

authors conclusions.

17. Immunostaining of cultured SCs for HOPX and JUN indicated that ICG inhibitor promoted maturation (Figure 6E) although no quantification was provided. Percentage of positive cells (and JUN/HOPX double positive cells) should be included. Many cells still seem positive for JUN with ICG, but this is an immature marker. Shouldn't numbers of JUN-expressing cells be decreased in those conditions?

18. Culture of mouse spermatogonial stem cells (GS cells) on SCs indicated that SC maturation correlated with enhanced GS cell growth (Figure 6). Colonies of GS cells are proposed to be larger when grown on healthy vs. iNOA SCs although this is not particularly evident from cell counts at day 9. This point should be reconciled and colony size quantified. Besides cell counts, do the GS cells express similar levels of stem-associated markers (Gfra1, Id4 etc) when co-cultured on SCs from the different sources? Importantly, spermatogonial stem cells in rodents are strongly proliferative and functional during postnatal testis development when SCs are immature. It doesn't therefore correlate that mature SCs in vivo are better at supporting spermatogonial stem cells than immature ones.

19. Dysfunctional SCs are proposed by the authors to be the cause of germ cell loss in NOA patients (Discussion). An alternative possibility, that germ cell loss then triggers aberrant changes in SCs is discounted, partially as the SCs in the different types of NOA patients (all with no germ cells) show different defects by gene expression pattern. However, as the detailed clinical history of these patients is unclear, is it not possible that germ cell loss at different developmental stages in the different NOA cases results in distinct responses from the SCs? Earlier loss causing maturation defects and later loss resulting in inflammation, for example? Other supporting arguments, that mouse GS cells do not drive SC maturation in vitro can be confounded by other effects, not least the species difference. Alternative explanations should be considered and this part of the discussion modified.

20. 9 cell clusters are identified by initial analysis of scRNA-seq data. However, Table S1 refers to 8 clusters and is marked as Table 1 in the text rather than S1. This should be corrected.

21. Occasional spelling errors are noted. Text should be checked carefully.

Reviewer #3:

Remarks to the Author:

Zhao et al. submitted the manuscript 'Single-cell atlas of human developing and azoospermia patients' testicles reveals the roadmap and defects in somatic microenvironment'.

This manuscript presents a highly comprehensive single cell RNA-Seq dataset obtained from 17 human testicular tissues and almost 90,000 cells that were profiled. Of the different testicular cells, the authors mainly focused on the somatic Sertoli cells that have been hardly covered by previous publications, due to the difficulty in obtaining sufficient cell numbers. Authors present highly interesting data on developmental aspects of the Sertoli cells and can identify 3 distinct stages of Sertoli cell maturation. What is more, they assess specifically selected patient samples (KS and Y chromosome microdeletions, iNOA) and can unveil maturation defects and associated pathways. Finally, authors even provide in vitro experiments performing Sertoli cell cultures and provide intriguing results, which however need to be further substantiated. The specific comments are outlined below:

As the authors focus 'only' on the Sertoli cells, the title, which refers to defects in the somatic microenvironment, seems inappropriate.

Comments regarding the results:

3.7% mitochondrial genes were detected per cell. Please indicate the range as there seem to be

Sertoli cells with much higher percentages.

Based on the low representation of Sertoli cells in published scRNA-Seq datasets, please provide the information on Sertoli cell numbers present in the different sample types.

'Total 1101 differentially expressed genes (DEGs) with a fold change of log₂ transformed UMI >1 of each cluster were identified.' Can you please specify the comparisons that were made for this. Was each cluster compared to the rest of the dataset?

Authors write 'our results suggest that the dysfunction of Sertoli cells contributes to the failure of the establishment of a suitable microenvironment in NOA.' Based on the data presented in Fig 1, this claim is not supported. How can the author's exclude that the Sertoli cells change their expression profile due to the absence of the germ cells?

Figure 1B: Please add number of samples in brackets- I assume it is adult (n=5)?

Figure S1B: Hormonal parameters for adults including AZFa Microdeletions and Klinefelter Syndrome, should be shown as individual data points. Datasets from the latter two groups should be very different from the normal.

Figure 3K: Authors report in the text, that 'nearly all Sertoli cells were HOPX-positive. This claim is not supported by the image the author's provide. It is unclear if the staining is even Sertoli cell-specific. Moreover, in the provided cross section, Sertoli cells appear to be negative in the remaining tubule?

Figure S5: The legend within A is very small and is unreadable for D. Please modify.

Figure 4E: The legend in the image is very difficult to read, please increase in size.

Figure 4F: The authors report that their finding 'was corroborated by HOPX levels that were lower in iNOA samples than in samples from other patients and healthy adults'. While it is very difficult to assess lower expression levels in the IF images that are provided, it appears as though the expression pattern has changed. The iNOA samples appear to show a nuclear rather than a cytoplasmic staining. Can the authors comment on this?

Figure S6G: The legend within the figure is too small to read, please increase in size.

What do the authors mean by 'over proliferation of interstitial cells were also observed' with regard to the iNOA patients? Do the authors mean a higher proportion of the interstitium and how was that evaluated?

Figure 6: Many groups have attempted and failed to isolate and culture human Sertoli cells. For this reason, authors need to provide more information on the isolation procedure and the characterization of Sertoli-cell cultures. Morphologically, cells shown in Fig 6B and H resemble SMA- positive peritubular cells. Ideally, authors could provide expression data including stable markers such as SOX9 and SMA. Alternatives could be functional data performing Sertoli cell stimulation with FSH. This is of importance for the quantitative data shown in C, D, G and I as these results greatly depend on comparable cell proportions in respective cell suspensions. With regard to this figure, authors write that 'the proportion of HOPX-positive cells in the ICG-treated group was significantly higher than that in the control group. However, the proportion of Jun showed an opposite trend and did not show any co-staining with HOPX (Figure 6E). However, in Fig. 6E, authors merely show pos. and neg. stained cells indicated by arrows/arrowheads. In case the authors generated quantitative data, which would justify the use of significantly more or less, the author's should refer and present this information.

In general, this manuscript would greatly benefit from a summarizing model, which should cover the normal developmental changes of Sertoli cells plus the alterations in the patient groups that have been analyzed.

Reviewer #1 (Remarks to the Author):

Response to Reviewer #1:

Thank you for your suggestions, in this study, we first focused on the normal development of Sertoli cells (Figure 2-3) and then described the pathological changes in NOA Sertoli cells (Figure 4-6). Different sample combinations during analysis may lead to variation in dimensionality reduction patterns, it may cause confusion to readers. Therefore, we calculated the dissimilarity with two methods to give a more credible evidence and quantitative results to describe the difference between clusters. In addition, we reconciled our sequence data with previous studies, the merged samples covered different races, families, and laboratories, but did not change our results and conclusions. In general, after modification according to your suggestions, our conclusions are more understandable and credible. The following content is our point-by-point responses and the main corrections in the revised manuscript. Thank you again!

Major comments:

Figure 2 A (left): Based on the UMAP plot, the distance between adult cells (yellow color) and stage_a (2-5 years) is shorter than the distance between Stage_a and stage_b.

But in Figure 4A, the adult cells are far away from cells of 2-5 years. Any explanation for this?

Response: The UMAP dimension reduction calculation in Figure 2A was processed base on the top 1000 different expressed genes (DEG) and the top 20 PCA within these normal Sertoli cells. But in Figure 4A, the top 1000 DEGs were from normal and NOA cells, these DEGs tend to reflect the difference between normal and disease but may ignored some information in the normal development process. This change may cause the different spatial distribution between Figure 2A and 4A. Therefore, the distance of the UMAP plot could not fully reflect the similarity/dissimilarity between clusters. In order to accurately describe the similarity/dissimilarity (distance) within different ages and within different stages, we calculated the Jaccard and Bray-curtis distance between each two ages and two stages. Briefly, the Jaccard and Bray-curtis distance between Stage_a and Stage_b was shorter than that between Stage_b and Stage_c (Figure R1.1A-B), and the distance was positively correlated with ages (Figure R1.1C-D).

Figure R1.1. The dissimilarity between Sertoli cells at different ages and stages. A greater value represent a greater dissimilarity.

(A-D) Bray-curtis or Jaccard distance between Sertoli cells at different stages or ages are shown on bubble diagram. The gradient of bubbles sizes and color indicates low to high Bray-curtis or Jaccard value.

The three stages are the fundamental finding in the manuscript. The pattern is very clear but hope can have more validations to confirm that these patterns are not due to data sampling bias (e.g., gaps in ages, population differences (e.g., family, geography differences)).

Response: Thank you for your suggestion, it is important to exclude the sampling bias. So, we downloaded the testicular single-cell seq data from the previous studies of two other separate laboratories (Cairns's and Wilkinson's)(Guo et al., 2020; Sohni et al., 2019) and merged them with our data. The pseudotime analysis of the merged data also showed clear three stages development pattern (Figure R1.2A and R1.2B). And the result, Stage_a cells gradually decrease and Stage_b cells gradually increase before puberty and Stage_c cells become dominant after puberty, was as same as our original data (Figure R1.2C and R1.2G). This result has been added in the Figure S3 of the revised manuscript.

Figure R1.2. The pseudotime analysis of our data reconciled with other published datasets. (A, B) The pseudotime analysis of the merged data. Cells were labeled with different color according to their stage and data source (ours: 2, 5, 8, 11, 17 and adult; Guo's: 7, 13, 14; Sohni's: 0 years) (correspond to Figure S3A and B) (C) The percentage of three stages at different ages. (corresponds to Figure S3C). (D, E) The pseudotime analysis of the data in this study. (F, G) The pseudotime analysis of the data in Guo's study (Guo et al., 2020).

Overall it is a nice work.

Minor comments:

The manuscript is overall well written but some expressions need to be changed.

E.g.,

“We have a superficial understanding that Sertoli cell development after birth is mainly divided into two stages: the immature and mature stage.”

A better way to express this can be something like “Although most studies divided Sertoli cell development after birth into two stages: the immature and mature, it is largely unknown whether

there exists any intermediate or transition states” -- a more positive way to acknowledge current limitations.

Response: Thank you, and we have revised this part according to your suggestion.

Reviewer #2 (Remarks to the Author):

Response to Reviewer #2:

Thank you for your in-depth and detailed comments. In the revised manuscript, all protein expression level and the difference between clusters were quantified. For the Wnt inhibition assay, we re-added a comparison of different inhibitors, because this result was not showed in the previous manuscript because of the limit of article length. To give corresponding in vivo experimental evidence, we also added the β -catenin staining of testis section at different ages and NOA. In addition, some conclusions lack of sufficient evidence to support them were modified, such as the cell cycle phase identification and induced Sertoli cells can better support GS cells proliferation. In general, three stage model of Sertoli cells maturation is a novel discovery and the induced maturation culture system based on this theory is a promising avenue for the possible treatment of iNOA patients, your comments make the logic of this study more rigorous and the conclusions more credible. We have made corrections according to the comments. The following content is our point-by-point responses and the main corrections in the revised manuscript. Thank you again!

Points to be addressed are below:

1. While the manuscript focus is on SCs and other somatic cells, the scRNA-Seq identifies germ cell populations from spermatogonia to spermatids. The authors perform some immunostaining to highlight stem (GFRA1) and differentiating spermatogonia (KIT) in testis samples from different ages (Fig S1). Multiple recent papers have delineated different spermatogonial subtypes in human testis by scRNA-Seq (e.g. from the groups of Bradley Cairns and Miles Wilkinson). Importantly, these studies find that the most primitive spermatogonia express low levels of GFRA1 and rather are marked by high levels of TSPAN33, UTF1, PIWIL4 etc. Given the many single cell studies now published, it would be informative if the authors can correlate their spermatogonia data to these other studies and check expression of other stem cell markers by immunostaining. Text should be modified to include these points.

Response: Thank you for your suggestion, to give more information about the germ cells, we divided the germ cells cluster into 14 subsets according to reported markers. Then, we added the IHC staining of two primitive spermatogonia specific markers (UTF1 and PLZF) and two markers of primitive and differentiated spermatogonia marker (SMS and TKTL1) in the Figure S1 of the revised manuscript.

Figure R2.1. the expression pattern of known and novel spermatogonia marker genes.

(A) Heatmap showed the scaled expression level of known and novel marker genes during spermatogenesis. (corresponds to Figure S1E).

(B) Immunofluorescence staining for UTF1 (SSC marker), PLZF (SSC marker), SMS (SSC and SPG marker), TKTL1 (SSC and SPG marker) and double staining of SYCP3 and γ H2AX (SPC markers) in testicular paraffin sections of each age, illustrating the spermatogenic maturity in each sample. The scale bar represents 20 μ m.

2. From UMAP analysis of healthy and NOA samples in Figure 1, it was concluded that “Sertoli cells showed greater difference in spatial distribution than other populations.” Eyeballing the figure does suggest this but this is a subjective statement. Can this be supported by quantitative analysis?

Response: Thank you for your suggestion, to give a more objective assessment of the difference, we used the top 1000 DEGs (the UMAP plots were also obtained base on these DEGs) to calculated the dissimilarity between normal adult and patients with two methods (Jaccard distance and Bray-curtis distance). The Jaccard and Bray-Curtis distance index clearly showed “the Sertoli cells showed greater difference” between different sample types. These results were added in the Figure 1 and S1 of the revised manuscript, the methods were also added in the method part of the revised manuscript.

Figure R2.2. The dissimilarity of somatic cells between different types of patients.

(A) Dissimilarity of somatic cells between normal adult and AZFa_Del, KS or iNOA are shown on bubble diagram. The gradient of bubbles sizes indicates low to high scaled Bray value, and the gradient of red indicates low to high scaled Jaccard values. (corresponds to Figure 1G).

(B) Dissimilarity of somatic cells between iNOA and AZFa_Del or KS are shown on bubble diagram. The gradient of bubbles sizes indicates low to high scaled Bray value, and the gradient of red indicates low to high scaled Jaccard values. (corresponds to Figure S1G).

3. From pseudotime analysis of healthy SCs (Figure 2), the authors conclude that these cells go through a defined developmental pathway (stage a to b to c) with increasing age. Confusingly, the pseudotime plot shows a bifurcation from a to c and a to b stages. Why is this? Interestingly, a recent study of human testis during puberty by the Cairns group (Guo et al. Cell Stem Cell 2020, 26:262) suggests that there are two immature SC states that converge into a single mature state. The authors need to reconcile their datasets with this previous study and clarify the developmental pathway supported by the data.

Response: Thank you for your question and suggestion. Base on the merged data (Figure R2.3)(Guo et al., 2020; Sohni et al., 2019), we still assumed two immature SC states are two continuous stages and the Stage_b SC were differentiated from Stage_a.

1) The percentage of Stage_a SCs count was decreased with age (Figure R2.3C). It means that most Stage_a cells underwent apoptosis or differentiation. The GO analysis of Stage_a cells did not find any apoptosis relative terms enriched but find “cell differentiation” and “maintenance of cell number”, furthermore, there was no report about an increase apoptosis of SC around puberty. So, the most likely explanation is that most Stage_a cells become Stage_b (differentiation) cells around puberty.

2) Another possibility is Stage_b cells proliferated rapidly around puberty (Figure R2.3C). In the 11 years old sample, the percent of Stage_b were higher than 80%. However, from this study, we know that Stage_b cells was not as proliferative as Stage_a. So, the increased percentage was not because of a higher proliferation of Stage_b cells but the differentiation of Stage_a cells.

However, we still need more experimental evidence for this hypothesis. We have added this discussion into the revised manuscript.

Figure R2.3. The pseudotime analysis of our data reconciled with other published datasets. (A, B) The pseudotime analysis of the merged data. Cells were labeled with different color according to their stage and data source (ours: 2, 5, 8, 11, 17 and adult; Guo's: 7, 13, 14; Sohni's: 0 years) (correspond to Figure S3A and B). (C) The percentage of three stages at different ages. (corresponds to Figure S3C). (D, E) The pseudotime analysis of the data in this study. (F, G) The pseudotime analysis of the data in Guo's study (Guo et al., 2020).

4. In Figure 2G the authors immunostain testis sections with a range of identified markers of the distinct SC stages to support the proposed developmental pathway. This dataset needs to be quantified and the significance of the GATA4 counterstain mentioned in the figure legend. The immunostained proteins have different localisation patterns – are these as expected? Some further details on these marker proteins should be provided.

Response: The quantification and significance of these stage specific marker genes (Figure R2.4) have been added in the Figure 2G and its legend in the revised manuscript. The percentage of JUN+ Sertoli cells was 63.2% in 2 years, 16.6% in 11 years and 1.3% in adult (Figure R2.4A and B). Unlike JUN staining, it is hard to distinguish positive and negative stain for ENO1 and

DEFB119. So, we analyzed the fluorescence intensity of these two proteins in per Sertoli cells, the statistics were matched with the RNA-seq data (Figure R2.4C and D). According to the Human Protein Atlas database, JUN located in nucleoplasm, and ENO1 located in nucleoplasm, plasma membrane and cytosol, and DEFB119 was in extracellular region or secreted, all these localization patterns were matched with the IHC staining results of this study. The description of subcellular location and detail information have been added in the relevant passages of the revised manuscript.

Figure R2.4. The expression pattern of stage specific genes.

(A) Immunofluorescence co-staining of GATA4 (red) with JUN (green, upper panel), ENO1 (green, middle panel), and DEFB119 (green, lower panel) in human testicular paraffin sections at three ages. The long arrow marks Sertoli cells with high expression level of these marker genes. The scale bar represents 20 μ m. *** p <0.001 (compared with normal adult). (corresponds to Figure S2F).

(B-D) The statistics of the percentage of JUN+ Sertoli cells (B), and the fluorescence intensity of ENO1(C) and DEFB119(D) in per Sertoli cells. (correspond to Figure S2G).

5. From GO analysis of DEGs of different SC stages (Figure S3), the authors highlight a selection of terms including “stem cell differentiation” and cell fate commitment” for stage “a” SCs. Other GO terms were also found but not mentioned e.g. “rhythmic processes” and “covalent chromatin modification”. It seems that the terms that suited the proposed developmental model were focused

on but other potentially important terms were ignored. This section should be discussed in more detail.

Response: Because of the limit of article length. We have selected some GO terms that are obviously related to development. However, other GO terms such as “rhythmic processes” and “covalent chromatin modification” may also show potential relationship with Sertoli cell development. For example, a previous study reported a significant diurnal variation in inhibin B, with peak values in the early morning and nadirs in the late afternoon, followed by gradual increasing nocturnal values (Carlsen et al., 1999). From the result of the new Figure S4F, Stage_a Sertoli cells expressed more inhibin B than Stage_c cells, indicated immature Sertoli cell may be more susceptible to **rhythm processes**. In addition, rhythm disruption is associated with **chromatin modification**, gene-expression regulation, and macromolecular metabolism (Möller-Levet et al., 2013). However, the effects of “**rhythm processes**” and “**chromatin modification**” on Sertoli cell development are rarely reported and need further study. Related discussions have been added in the revised manuscript

6. Mitotic phase prediction of different SC stages (Figure S3) was performed and stage “a” cells were concluded to be most proliferative. However, no significance value is provided with this analysis. Are the differences significant?

Response: From the heatmap, we found the expression level of S and G2M phase genes in Stage_a were obviously higher than Stage_b and c (Figure R2.5A). However, for those cells who don't proliferate rapidly, transcriptome-based bioinformatic analysis is difficult to accurately define the cell cycle phase of a cell. Cell cycle prediction of SCs was processed with the Seurat package in R. In the analysis processes, all cells were thought in the cell cycle by default. This method may not work well for mature Sertoli cell which were not proliferation in vivo. So, we deleted the Figure 3F in the previous manuscript and used the expression level quantification of these cell cycle phase specific genes instead (Figure R2.5B). Although we changed the method to describe the proliferation of three stages SCs, the main conclusion that Stage_a is most proliferative did not change, and we also checked this result in protein level with KI67 and PCNA stain (Figure R2.5C).

Figure R2.5. the expression level of cell cycle genes during Sertoli cell maturation.

(A) Heatmap of cell cycle-specific genes in Sertoli cells at each stage. (corresponds to Figure S3H).

(B) The expression level of cell cycle-specific genes in Sertoli cells at each stage. (corresponds to Figure S3I).

(3) Immunofluorescence co-staining of GATA4 (red) with KI67 (green, upper panel) and PCNA (green, lower panel) in human testicular paraffin sections at three ages. The long arrow mark Sertoli cells with KI67/PCNA and GATA4 double positive cells and the triangular arrow mark KI67/PCNA positive but GATA4 negative cells. (data not shown in manuscript).

7. IPA analysis of SCs suggests potential pathways important at different developmental stages (Figure 3). The authors highlight a limited number of pathways in the text and refer to the others as “et al”. Other identified pathways have seemingly been ignored/de-emphasized but the rationale for this is unclear. This same point applies to identification of transcription factors important for each stage on the next page. The text should be revised to better describe the available data. What is the relevance of “cardiac hypertrophy signaling” for stage “b” cells and why does this indicate that structural remodeling mainly occurs at that point?

Response: Because of the length limit of article, we only discussed some terms that are directly related to the characteristics of each stage of Sertoli cell development. Expect for IPA terms that we highlight in the previous manuscript, other pathways also have a potential effect on Sertoli cell

development. For example, 1) “**Unfolded protein response**” enriched in the early two stage may be because immature Sertoli cells lack organelles such as the endoplasmic reticulum (Hess and França, 2005). 2) Rho relative signaling pathways (“**RhoA Signaling**” and “**Regulation of Actin-based Motility by Rho**” in Stage_b and “**RhoGDI signaling**” in Stage_c) promote reorganization of the actin cytoskeleton and regulate cell shape, attachment, and motility (Hall, 1998). During the development process, Sertoli cells migrate from lumen to the region near the basement membrane, and their uniform, small and round nuclei shape change to irregular, large, and tripartite shape, in addition, mature Sertoli cells also feature higher cytoplasm ratio and extensive cytoplasmic projections that encircle the germ cells. All above change occurred around puberty (Stage_b cells dominated in count), considering Rho relative signaling pathways up-regulated since Stage_b, we assume the structural remodeling mainly occurs in Stage_b and Rho related pathway is a candidate regulator for this process. 3) VDR/RXR/PPAR relative signaling pathways (“**VDR/RXR Activation**” and “**PPAR Signaling**” in Stage_b, and “**PPAR α /RXR α Activation**” and “**Superpathway of Cholesterol Biosynthesis**” in Stage_c) regulate the cholesterol and lipid metabolism, which is an important part of SCs maturation and associated with steroid hormone production in SCs. 4) Cardiac hypertrophy process also depend on reorganization of the actin cytoskeleton and structural remodeling. Many studies reported cardiac hypertrophy and cardiac cell remodeling were regulated by HOPX-WNT pathway (Bergmann, 2010; Schneider et al., 2015), which was also associated with Sertoli cell maturity. Considering mesoderm-derived cells are relatively conservative in regulation, we thought “**Cardiac Hypertrophy Signaling**” should be added in the manuscript as a reference for further studying Sertoli cell maturity. All the detail description have been added in the result and discussion part of the revised manuscript.

8. For immunostaining data of Figure 3K, how many tubule cross-sections were scored? This information needs to be included in the figure legend and/or text.

Response: The tubule count has been added in the new Figure 3K. Because the size of 2 years old testis tissue was small, only 6-8 tubule in this sample were counted.

9. From analysis of ligand-receptor interaction between SCs and spermatogonia, “most” pairs involved TGF- β signalling (Figure S4). The authors should indicate in the text the percentage of pairs. Was GDNF-GFR α 1/Ret signalling identified? Are any FGFs identified? Rodent studies indicate key roles for both pathways in spermatogonial self-renewal. On a related note, Figure S4F is not described in the text. What is the relevance of these panels?

Response: Sorry for that mistake, “most” is not an inaccurate description here. We means TGF- β signaling relative pairs in Stage_a (6/55), in Stage_b (4/28), in Stage_c (2/16) and 12/99 in total, it is the highest percentage among all pairs. The expression of GDNF was low in all samples, so the CellPhone analysis did not identify any significant GDNF-GFR α 1/Ret signaling and FGFs relative pair. It may be because the sequencing depth of single cell was not enough. However, considering the importance of GDNF and FGF in spermatogenesis, we also added the expression pattern of these two signaling in the revised manuscript (Figure R2.6B and C). In addition, EPHA4/TIMP1-FGFRs pairs were detected in Stage_c (Figure R2.6A). The previous Figure S4F (the new Figure S4I) gave a protein expression level evidence showed that Sertoli cell could interact with both germ cells and other somatic cells via TGF- β signaling (INHA-ACVR2B and

INHA-TGFBR3) and the expression level of INHA protein was similar to the RNA seq data. In addition, the description of the Figure S4F (the new Figure S4I) has been added in the results part of the new manuscript.

Figure R2.6. Dynamic changes of the interactions between Sertoli cells and other testicular cells. (A) Heatmap showing the matching strength of the interaction between Sertoli cells and other testicular cells. (corresponds to Figure S4A). (B, C) Heatmap showing the spatial expression pattern of ligands and receptors of the GDNF-GFRA1/RET and FGFs-FGFRs signaling. (corresponds to Figure S4G and H).

10. Cell cycle prediction of SCs from iNOA patients indicated enhanced proliferation (Figure 4C). Is this change in proliferation vs healthy individuals significant? Confusingly, while percentage of iNOA SCs predicted to be in S-phase was increased, those in G2/M were decreased and those in G0/G1 similar to controls. Can this also suggest that iNOA cells are arrested in S-phase? If cells are cycling more it might be predicted that both S and G2/M phases are increased while G1/G0 cells decreased. This point should be clarified. On a related note, most metabolic pathways are predicted to be downregulated in iNOA SCs but these cells are concluded to be more active/proliferative. Given the energy demands of cell cycle, this conclusion seems counter-intuitive.

Response: Thank you for your suggestion and correction. With the same reason of comment 6, we use the absolute expression level but not the percentage of S and G2M specific genes to describe the ability of cell proliferation (Figure R2.7A and B). The S and G2M genes expression level in iNOA Sertoli cells were significantly higher than that in normal adult, indicated iNOA Sertoli

cells were more proliferative (Figure R2.7A). As for the metabolic analysis, we checked the code and found the oxidative phosphorylation analysis in old Figure S4D was based on a mistake gene list, the corrected result was added in the new manuscript. Furthermore, the energy metabolism is proportional to the transcriptional level, so we used the expression level of energy metabolic genes to represent the metabolism level, and used the percentage to represent the major metabolic pathways in the revised manuscript (Figure R2.7C-D and F). The absolute expression level of energy metabolic genes in iNOA were also higher than normal (Figure R2.7E and F), it matched the cell cycle result of iNOA cells.

Figure R2.7. cell proliferation and energy metabolism analysis.

(A, B) the expression level of S and G2M phase specific genes in Sertoli cells under pathological state (A) and three stages (B). (corresponds to Figure 4C and S3I).

(C) the expression level of energy metabolic genes in nine clusters of testicular cells (left panel) and three stage of Sertoli cells (right panel). (corresponds to Figure S3J).

(D) the percentage of energy metabolic genes in three stage of Sertoli cells. (corresponds to Figure S3K).

(E, F) the expression level of energy metabolic genes in normal Sertoli cells at different ages (D) and in iNOA (E). (corresponds to Figure 4D and 5E).

11. From HOPX immunostaining in Figure 4F it is concluded that iNOA SCs show reduced expression of this maturation marker but quantification is not provided. The percentage of SCs positive for this marker should be included.

Response: Thank you for your suggestion. Unlike the expression pattern in children, we found almost all adult Sertoli cells (GATA4 positive cells in normal adult and NOA patients) expressed HOPX at some level. So, it is difficult to compare the percentage of SCs positive for HOPX. Considering the HOPX was mainly expressed in nucleoplasm (Figure R2.8A and B), as same as GATA4. We selected the GATA4 positive region and calculated the average fluorescence intensity of HOPX staining in each GATA4 positive region as the expression level of HOPX protein (Figure R2.8D). This method was also used to calculate the expression level of β -catenin in the new Figure S5 and S8. The statistical results were shown in the new Figure 4G

and the detail of the method was also added in the method part of the revised manuscript.

Figure R2.8. Expression level of HOPX in normal adult and three types of pathological Sertoli cells.

(A) Immunofluorescence co-staining of GATA4 (red) and HOPX (green) in normal adult and three types of Sertoli cells. The scale bar represents 5 μm .

(B) The localization of HOPX in Sertoli cells according to COMPARTMENT database.

(C) The statistics of cell nucleus located HOPX expression is shown as histogram (G). ** $p < 0.01$, *** $p < 0.0001$ (compared with normal adult).

(D) Cell nucleus regions were determined according to GATA4 staining (yellow circle region), and then HOPX fluorescence intensity was collected for statistics.

12. At multiple points, the authors refer to the “Johnsen” score of testis pathology. For the benefit of general readers, the meaning of the scores should be highlighted in the text e.g. no germ cells present.

Response: Thank you for your suggestion. We have added a more detail description for the testis pathology in the new manuscript.

13. The authors state that low expression of UTY was detected in AZFa-del SCs suggesting that multiple copies of this gene are present outside this deleted Y chromosome region. Presumably this must be known? Can the authors elaborate on this point?

Response: Sorry for that oversight. All genes AZFa region were single copy. The AZFa_Del patient were diagnosed by a complete deletion sY84 and sY86 site(Krausz et al., 2014). However, these two sites don't cover UTY gene (located in chrY 13230770-13480670)(Tang et al., 2020), so this patient still have the UTY gene. We have revised the mistake of relevant parts in the new manuscript.

Figure R2.9 The localization of AZFa gene and test site in Y chromosome.

14. Detailed analysis of iNOA SCs (Figure 5) suggests different maturation states which are then compared to stages a,b and c from analysis in Figure 4B. The different iNOA SC stages are concluded to “resemble” stages a and b. However, in Figure 4B these cells have already been plotted onto stage a and b regions of the pseudotime trajectory so presumably are identifiable as these stages. Can the authors clarify the relationship between analysis in Figure 4 and 5?

Response: Thank you for your question. According to the Figure 4B, iNOA SC could be identified as Stage_a and Stage_b, it means the different iNOA SC stages are concluded to “resemble”

stages a and b. This result was calculated with monocle package in R base on top 1000 variable genes, it could not reflect all the characteristics of iNOA SC in Stage_a or Stage_b, for example, whether other genes in Stage_a iNOA SC were different form normal Stage_a SC? Based on biological background and disease cognition, there must be some difference between iNOA SC and normal immature SC in both Stage_a and Stage_b at some level. In the old Figure 5D, we calculated the DEGs in each cluster and compared them with DEGs of each normal Stage, but this result may cause confuse to readers, so we used a new method to exhibit the difference between iNOA cells and normal in same stage. In the new method, we first identified the genes with mean expression greater than 0.1 UMI in each stage, and then calculated the percentage of DEGs between iNOA and other normal Sertoli cells in the same stage (the gene with fold change less than 0.25 were named as overlap genes). A higher percentage of overlap genes represent a higher similarity between iNOA and other cells in the same stage. We found 7854(78.77%) overlap genes in Stage_a and 973(23.35%) in Stage_b, indicated the Stage_a iNOA cells was more similar to the normal cells at this stage, but the Stage_b iNOA cells showed greater difference/pathology. The 822(14.32%) DEGs in Stage_c come from the difference between normal adult cells and other Stage_c cells (11 and 17 years) (Figure R2.10C). This results and analysis method have been added in the revised manuscript.

Figure R2.10 the maturation states of Sertoli cells in normal adult and iNOA.

(A) Normal adult and iNOA Sertoli cells are highlighted red in the UMAP plot (upper row) and the pseudotime trajectory plot (lower row). (corresponds to Figure 4B).

(B) UMAP plot showing the heterogeneity among iNOA Sertoli cells; a clear difference was observed between iNOA and normal adult Sertoli cells. (corresponds to Figure 5A and B).

(C) The ratio of different expressing genes (DEGs) and overlap genes between Stage_a/b/c and cluster 1/2/3. (corresponds to Figure 5D).

15. From a culture-based assay of iNOA SCs, it is concluded that these cells are more proliferative than healthy SCs (Figure 5H). However, the difference in growth rate, although statistically significant, appears very mild. Would this change in proliferation rate be biologically significant? On a related note, further details of how this in vitro assay was performed should be included in the text.

Response: The proliferation of iNOA SCs were more active than normal adult, the difference is mild. It may be because (1) both normal adult and iNOA SCs dedifferentiate and re-entry to an immature state after isolated from *in vivo* to *in vitro*; (2) Sertoli cells were big in size and fibroblast-like in shape *in vitro*, so they grow slowly and are more prone to contact inhibition, this may also narrow the differences between OA and NOA SC proliferation. However, the difference is significant in statistics, and the expression level of cell cycle specific genes in iNOA SCs were also higher than normal adult. Therefore, we believe this result is biologically significant.

The proliferation assay was determined using a Cell Counting Kit (Dojindo, CK04-01), 3000 normal or iNOA Sertoli cells were seeded in 96-well plates and starved (DF12 medium without FBS) for 16 h. Then replace the medium with 10% FBS DF12 medium, change the medium every 2 days. Before test, CCK-8 solution was added to the cells and incubated for 3 h. The absorbance at 450 nm (A450 nm) was measured with a Sunrise microplate reader (TECAN). Each group each day had 5 repeats, the difference was calculated using nonparametric test with GraphPad Prism 8.0.2. This description has been added in the method part of the new manuscript.

16. A role of the Wnt pathway in SC maturation is concluded from analysis of scRNA-seq data and treatment of cultured SCs with Wnt pathway inhibitor ICG (Figure 6). Is the ICG specific to the Wnt pathway or are there expected off-target pathways? ICG is reported to bind the common coactivator CBP and some effects on cells may be independent of effects on the Wnt pathway (e.g. Arensman et al., PMID: 25082960). Are classical Wnt target genes affected in SCs by treatment with ICG? What does NC refer to in the figures? On a related note, studies in rodents have demonstrated that constitutive Wnt/b-catenin signalling in SCs blocks their maturation (Tanwar et al. *Biology of Reproduction* 2010, 82: 422–432) – in line with the authors conclusions. However, this study is not mentioned. This study should be highlighted in the text and compared to the authors conclusions.

Response: Thank you for your question. When chose Wnt pathway as the target to induce SCs maturation, we had compared the effect of three common Wnt pathway inhibitors: ICG-001, XAV939 and KY02111 (the concentrations were all 10uM which were much higher than the IC50 concentrations of these inhibitors). We found all inhibitors mildly changed the total expression level of β -catenin protein. The β -catenin protein would entry into nuclei when Wnt pathway was activated, and the ICG-001 treated group showed a lowest level of nuclei located β -catenin (Figure R2.11A), in addition, ICG-001 also showed a stronger inhibition for SCs proliferation than other inhibitors (Figure R2.11B), these results were deleted in the previous study because of the limit of article length. So, we chose ICG-001 to processed the further function and mechanism experiment. According to Arensman's study, ICG-001 significantly inhibited *in vitro* and *in vivo* PDAC growth by inducing G1 cell-cycle arrest through effects that were largely decoupled from its activity as a Wnt inhibitor. This result was similar with ours: ICG-001 inhibit the proliferation of SCs and reduced the expression of Cyclin D1. However, more and more studies reported the role of β -catenin in promoting cell proliferation through cooperation with CBP(Cui et al., 2019; Jiang et al., 2015; Yu et al., 2017), indicated ICG-001 induced cycle arrest was not Wnt/ β -catenin independent. Furthermore, other function change of ICG treated SCs couldn't explain only by Wnt/ β -catenin independent CBP induced cell cycle arrest, such as the increasing of HOPX, the decreasing of INHA and AMH, and the change of some classical Wnt

target genes (Figure R2.11C). Tanwar's study showed that constitutive WNT/ β -Catenin Signaling in Sertoli cells leads to their continuous proliferation and compromised differentiation, resulting in increased germ cell apoptosis and infertility (Tanwar et al., 2010). According to their result, the increased viable GS cells count in ICG treated group may be because of a decreased apoptosis with or without an increased proliferation. We have added Tanwar's study and revised our relative conclusion and discussion in the new manuscript.

Figure R2.11 the inhibition of Wnt/ β -catenin pathway in Sertoli cells in vitro.

(A) Immunofluorescence staining of β -catenin with or without nucleus translocation in Sertoli cells showed the activated or inactivated Wnt/ β -catenin pathway. (corresponds to Figure S8A).

(B) the Sertoli cell count after treated with different Wnt/ β -catenin pathway inhibitors for 5 days. (corresponds to Figure S8B).

(C) the fold change of classical Wnt target genes in Sertoli cells after ICG treatment. (corresponds to Figure S8D).

17. Immunostaining of cultured SCs for HOPX and JUN indicated that ICG inhibitor promoted maturation (Figure 6E) although no quantification was provided. Percentage of positive cells (and JUN/HOPX double positive cells) should be included. Many cells still seem positive for JUN with ICG, but this is an immature marker. Shouldn't numbers of JUN-expressing cells be decreased in those conditions?

Response: The percentage of JUN positive cell decreased from 19.02%/19.66% (normal adult/iNOA) to 13.53%/12.82% (normal adult/iNOA) and the difference was significant (Figure R2.12B). We will improve this maturation induction culture system in the further study to increase its efficiency. The quantification and its relative method have been added in the revised manuscript.

Figure R2.12. The change of JUN and HOPX in Sertoli cells after ICG treatment.

(A, B) Immunofluorescence co-staining of JUN (red) and HOPX (green) in cultured iNOA and normal adult Sertoli cells with or without ICG treatment (A). The statistics of JUN/HOPX positive cells count is shown as histogram (B). The scale bar represents 20 μm . * $p < 0.05$, ** $p < 0.01$, *** $p < 0.001$ (compared with DMSO treated group). (corresponds to Figure 6F and G).

18. Culture of mouse spermatogonial stem cells (GS cells) on SCs indicated that SC maturation correlated with enhanced GS cell growth (Figure 6). Colonies of GS cells are proposed to be larger when grown on healthy vs. iNOA SCs although this is not particularly evident from cell counts at day 9. This point should be reconciled and colony size quantified. Besides cell counts, do the GS cells express similar levels of stem-associated markers (Gfra1, Id4 etc) when co-cultured on SCs from the different sources? Importantly, spermatogonial stem cells in rodents are strongly proliferative and functional during postnatal testis development when SCs are immature. It doesn't therefore correlate that mature SCs in vivo are better at supporting spermatogonial stem cells than immature ones.

Response: Thank you for your suggestion. The colony size quantification was similar with the result of GS cells count (Figure R2.13B and C). The expression level of stem-associated markers were measured with QPCR, the change was mild (within 1.5-fold change) and not significant (Figure R2.13A).

SSCs in rodents are strongly proliferative and functional during postnatal testis development when SCs are immature, however, fundamental differences exist between human and rodent with respect to hormonal control of puberty and onset of spermatogenesis, humans lack the equivalent of this first wave of spermatogenesis that in rodents and instead are believed to maintain spermatogonia in an silent state prior to the initiation of puberty (Guo et al., 2020). During puberty, most Sertoli cells have entry into Stage_b and some cells in Stage_c (8-14 years). What's more, from Tanwar's study and the response for your 16th question, we realized that the increased viable GS cells count in ICG treated group may be not only caused by a decreased proliferation but also an increased apoptosis, because matured SCs were able to better support SSC cell survive (Tanwar et al., 2010). We discussed the result from both apoptosis and proliferation aspects in the revised manuscript, however, considering the difference between human and rodents may be caused by spermatogonia themselves or somatic spermatogenic microenvironment, using human SSC in future research could better address this problem.

Figure R2.13. GS cells co-cultured with ICG induced Sertoli cells

(A) Expression change of SSC marker genes in GS cells co-cultured with Sertoli cells. Sertoli cells were first treated with or without ICG for 14 days. (corresponds to Figure S8E).

(B, C) The colony size of cultured GS cells using iNOA and normal adult Sertoli cells with or without ICG treatment as feeder cells. The colonies are surrounded by blue lines (B). The statistics of colony size is shown as histogram (C). * $p < 0.05$, *** $p < 0.001$ (compared with DMSO treated group). (corresponds to Figure S8F and G).

19. Dysfunctional SCs are proposed by the authors to be the cause of germ cell loss in NOA patients (Discussion). An alternative possibility, that germ cell loss then triggers aberrant changes in SCs is discounted, partially as the SCs in the different types of NOA patients (all with no germ cells) show different defects by gene expression pattern. However, as the detailed clinical history of these patients is unclear, is it not possible that germ cell loss at different developmental stages in the different NOA cases results in distinct responses from the SCs? Earlier loss causing maturation defects and later loss resulting in inflammation, for example? Other supporting arguments, that mouse GS cells do not drive SC maturation in vitro can be confounded by other effects, not least the species difference. Alternative explanations should be considered and this part of the discussion modified.

Response: Thank you for your suggestion. It is a very interesting hypothesis. In the future, if we being able to collect testicular samples from model animal and children who were diagnosed with AZF deletion, KS or SCOS (Sertoli Cell Only Syndrome) would help to investigate the hypothesis. We have added this possibility in the discussion part of the revised manuscript.

20. 9 cell clusters are identified by initial analysis of scRNA-seq data. However, Table S1 refers to 8 clusters and is marked as Table 1 in the text rather than S1. This should be corrected.

Response: this mistake have been corrected.

21. Occasional spelling errors are noted. Text should be checked carefully.

Response: All text spelling have been checked again by two researchers and the spelling errors have been corrected.

Reviewer #3 (Remarks to the Author):

Response to Reviewer #3:

Thank you for your comments, our study benefit a lot from your suggestion. We captured more Sertoli cells than previous study for RNA sequence, based on the improvement of sample processing and more sampling at different ages and patients, it provided a chance to intensively study the maturation of Sertoli cells and the somatic spermatogenic microenvironment. In the revised manuscript, (1) all protein expression level and the difference between different cell clusters were quantified; (2) details of the experimental method such as the isolation, culturing, and identification of Sertoli cells were added; (3) some small figure legends has been increased in size for reading. In addition, some supplementary experiments such as the inhibition of Wnt/ β -catenin pathway increased the reliability of our conclusions. In general, the revised manuscript better elucidated the maturation of Sertoli cells and the potential applications of this study. We have made corrections according to the comments. The following content is our point-by-point responses and the main corrections in the revised manuscript. Thank you again!

As the authors focus 'only' on the Sertoli cells, the title, which refers to defects in the somatic microenvironment, seems inappropriate.

Response: Thank you very much for the reviewers' suggestions. Our title is based on the following considerations: 1) unbiased analysis of single-cell sequencing includes all types of testicular cells; 2) We cover the interaction analysis of various types of testicular somatic cells, and we believe that they together constitute the microenvironment of spermatogenesis. 3) Sertoli is an important part of the testicular microenvironment and is inseparable from other components. Although the focus of our article analysis is Sertoli cells, we still emphasize the importance of microenvironment.

Comments regarding the results:

3.7% mitochondrial genes were detected per cell. Please indicate the range as there seem to be Sertoli cells with much higher percentages.

Response: Thank you for your suggestion. The range of mt-percent in all cells was from 0.00% to 39.97%, the mean value was $3.73 \pm 9.51\%$ (median: 3.42; 1st Qu: 0.7496; 3rd Qu: 9.8119) . This range in Sertoli cells was from 0.00% to 39.82%, the mean value was $12.59 \pm 10.93\%$ (median: 8.81; 1st Qu: 2.39; 3rd Qu: 22.28) .

Based on the low representation of Sertoli cells in published scRNA-Seq datasets, please provide the information on Sertoli cell numbers present in the different sample types.

Response: Thank you for your suggestion. The number of Sertoli cells in each sample has been added in the new Figure S1A of the revised manuscript. We speculate higher efficiency of Sertoli cells capture in this study because of these reasons:

(1) Larger size, irregularly shaped Sertoli cells may be more susceptible to cryopreservation damage. Unlike previous studies(Guo et al., 2020; Shami et al., 2020; Sohni et al., 2019), all testis

tissues in this study were isolated to single cells and performed cDNA library preparation within 12 hours of removal by surgery, and without any frozen process (tissue stored in 4°C before isolation in this study).

(2) We use “one step enzymatically digested” with 4 mg/ml collagenase type IV, 2.5 mg/ml hyaluronidase, and 1 mg/ml trypsin at 37°C for 20 min instead of the standard two step enzymatic isolation (Guo et al., 2020) or single collagenase type IV digestion(Sohni et al., 2019; Wang et al., 2018). In addition, given the risk of damaging certain types of big cells (such as Sertoli cells), cut-up of the tissue was avoided during this process.

A

Sample ID	Group	Tech of single-cell capture	Total Cell Number	Sertoli Cell Number
LZ011	2 years	10X Genomics	7886	4793
LZ009	5 years	10X Genomics	4239	1744
LZ005	8 years	10X Genomics	7515	2046
LZ008	11 years	10X Genomics	5253	631
LZ016	17 years	10X Genomics	5007	834
LZ003/LZ007	23/28 years (OA)	10X Genomics	6342/8840	211/856
LZ013/LZ014/LZ015	25/28/31 years (OA)	BD Rhapsody	3197/3467/3530	408/27/89
LZ002	31 years (AZFa Microdeletions)	10X Genomics	5203	435
LZ004/LZ010/LZ012	26/27/29 years (Klinefelter's Syndrome)	10X Genomics	2743/2740/2244	117/115/30
LZ017/LZ018/LZ019	32/26/31 years (NOA)	BD Rhapsody	7402/6526/6579	992/682/974

Figure R3.1. the information of enrolled samples. (corresponds to Figure S1A).

‘Total 1101 differentially expressed genes (DEGs) with a fold change of log2 transformed UMI >1 of each cluster were identified.’ Can you please specify the comparisons that were made for this. Was each cluster compared to the rest of the dataset?

Response: Yes, when calculating the DEGs in each cluster, we compared each cluster with the rest of other clusters. This method was also used in calculating DEGs in each Sertoli cells stage. However, in Figure 3D, 3E, 5J and 5K, the GESA analysis were based on the DEGs which come from comparison between the two corresponding groups. The detail of the analysis method has been added to the result and method part of the new manuscript.

Authors write ‘our results suggest that the dysfunction of Sertoli cells contributes to the failure of the establishment of a suitable microenvironment in NOA.’ Based on the data presented in Fig 1, this claim is not supported. How can the author’s exclude that the Sertoli cells change their expression profile due to the absence of the germ cells?

Response: Thank you for your suggestion. To exclude the effect of germ cells, we also compared iNOA with other two types of NOA (AZFa and KS), and found Sertoli cells also showed the greatest changed among somatic cells, indicating that different etiologies did cause different changes of Sertoli cells in the absence of germ cells. Furthermore, many studies have reported that conditional activated or deletion allele of some genes in Sertoli cells caused the changes of spermatogenic environment and damage to germ cells(Steger et al., 1999; Tanwar et al., 2010). However, all these results could not provide direct evidence of a causal relationship between SC changes and spermatogenesis disorders in this study. So, we have revised the conclusion in this paragraph to “the dysfunction changes of microenvironment in NOA patients are mainly manifest in Sertoli cells” in the new manuscript.

Figure 1B: Please add number of samples in brackets- I assume it is adult (n=5)?

Response: the number have been added in the Figure 1B, this information was also showed in Figure S1A of the revised manuscript (Figure R3.2).

Figure R3.2 the UMAP plot of testis cells at different ages. (corresponds to Figure 1B).

Figure S1B: Hormonal parameters for adults including AZFa Microdeletions and Klinefelter Syndrome, should be shown as individual data points. Datasets from the latter two groups should be very different from the normal.

Response: The hormonal parameters of normal adult and three NOA have been added in the Figure S1C of the revised manuscript (Figure R3.3).

Figure R3.3. The hormonal parameters of normal adult and three NOA types in this study. (corresponds to Figure S1C).

Figure 3K: Authors report in the text, that ‘nearly all Sertoli cells were HOPX-positive. This claim is not supported by the image the author’s provide. It is unclear if the staining is even Sertoli cell-specific. Moreover, in the provided cross section, Sertoli cells appear to be negative in the remaining tubule?

Response: We provided a higher quality image for the new Figure 3K of the revised manuscript (Figure R3.4A), this figure clearly showed a co-stained of HOPX and GATA4 (Figure R3.4B), the percentage of HOPX⁺/GATA4⁺ cells in all GATA⁺ cells also showed in the figure and result part (Figure R3.4C).

Figure R3.4. the immunofluorescence staining of HOPX and EGR3 in Sertoli cells at three ages. (A) Immunofluorescence co-staining of GATA4 (red) with a master regulator of Stage_a, EGR3 (green, upper panel), and a master regulator of Stage_c, HOPX (green, lower panel), in human testicular paraffin sections at three ages. The scale bar represents 20 μ m. *** p <0.001 (compared with normal adult). (corresponds to Figure 3J). (B) Magnification of GATA4 (red) and HOPX (green) stain in adult testicular paraffin sections. (C) The statistics of the percentage of EGR3+ or HOPX+ Sertoli cells (corresponds to Figure 3K).

Figure S5: The legend within A is very small and is unreadable for D. Please modify.

Figure 4E: The legend in the image is very difficult to read, please increase in size.

Response: The legends in these figures have been modified.

Figure R3.5. the revised figures for Figure S5A (Figure R3.5A) and 4E (Figure R3.5B).

Figure 4F: The authors report that their finding ‘was corroborated by HOPX levels that were lower in iNOA samples than in samples from other patients and healthy adults’. While it is very difficult to assess lower expression levels in the IF images that are provided, it appears as though the expression pattern has changed. The iNOA samples appear to show a nuclear rather than a cytoplasmic staining. Can the authors comment on this?

Response: Thank you for your question. HOPX staining in testis tissue has a nucleus location pattern (Figure R3.6A). Furthermore, according to uniprot database, both HOPX and Sertoli cell marker GATA4 were mainly expressed in nucleus (Figure R3.6B). HOPX prevents SRF-dependent transcription either by inhibiting SRF binding to DNA or by recruiting histone deacetylase (HDAC) proteins that prevent transcription by SRF, indicated functional HOPX was mainly located in nucleus. So, we selected the GATA4 positive region and calculated the average fluorescence intensity of HOPX staining in each GATA4 positive region as the expression level of HOPX protein (Figure R3.6D). The quantification was showed in the new Figure 4G of the revised manuscript (Figure R3.6C).

Figure R3.6. Expression level of HOPX in normal adult and three types of pathological Sertoli cells.

(A) Immunofluorescence co-staining of GATA4 (red) and HOPX (green) in normal adult and three types of Sertoli cells. The scale bar represents 5 μ m.

(B) The localization of HOPX in Sertoli cells according to COMPARTMENT database.

(C) The statistics of cell nucleus located HOPX expression is shown as histogram (G). ** $p < 0.01$, **** $p < 0.0001$ (compared with normal adult).

(D) Cell nucleus regions were determined according to GATA4 staining (yellow circle region), and then HOPX fluorescence intensity was collected for statistics.

Figure S6G: The legend within the figure is too small to read, please increase in size.

Response: The legends of S6A and S6G have been increased in size in the new manuscript.

Figure R3.7 the revised figures for Figure S6A (Figure R3.7A) and S6G (Figure R3.7B).

What do the authors mean by ‘over proliferation of interstitial cells were also observed’ with regard to the iNOA patients? Do the authors mean a higher proportion of the interstitium and how was that evaluated?

Response: The “over proliferation of interstitial cells” means a higher proportion of the interstitium, it is a common pathological change in NOA patients(Esteves, 2015; Goluža et al., 2014). It maybe because the lack of germ cells lead to an increased level of gonadotropins feedback in these patients. It could be evaluated by the ratio of cell count between within seminiferous (Germ cells and Sertoli cells) and within interstitium tubules (other somatic cells). As shown in the Figure R3.9, from five random field under 40X magnification field, we calculated the percent of interstitium cells in normal was 8% (254/3164) and in iNOA was 59% (176/297) (Figure R3.9A). This result was also supported by the cell count of our sequence data, in 5 adult and 3 iNOA samples, the percent of interstitium cells in normal was 15% (3640/23958) and in iNOA was 87% (18179/20865) (Figure R3.9B).

Figure R3.8. the ratio of cell count between interstitium and seminiferous tubules.

(A) Cell count statistics according to PAS staining.

(B) Cell count statistics according to single-cell sequencing data. Interstitium include Leydig_cells, PTM_cells, endotheliocytes, macrophages and VSM_cells, tubules cells include germ cells and Sertoli cells.

Figure 6: Many groups have attempted and failed to isolate and culture human Sertoli cells. For this reason, authors need to provide more information on the isolation procedure and the characterization of Sertoli-cell cultures. Morphologically, cells shown in Fig 6B and H resemble SMA- positive peritubular cells. Ideally, authors could provide expression data including stable markers such as SOX9 and SMA. Alternatives could be functional data performing Sertoli cell stimulation with FSH. This is of importance for the quantitative data shown in C, D, G and I as these results greatly depend on comparable cell proportions in respective cell suspensions.

Response: we have uploaded the protocol of human adult Sertoli cell isolation and culture to the Protocol Exchange, the protocol DOI is 10.21203/rs.3.pex-997/v1. The detail method was also described in the revised manuscript. The IHC staining of SOX9 and SMA (Figure R3.9A) showed the proportion of Sertoli cells was not significantly changed after ICG treatment.

Figure R3.9. Identification of human primary Sertoli Cells after treated with ICG-001 for 14 days. Sertoli Cells were stained with SOX9 (red) and SMA (green), and the percentage of SOX/SMA positive cells was noted according to three independent fields at 20X magnification. (corresponds to Figure S8C).

With regard to this figure, authors write that ‘the proportion of HOPX-positive cells in the ICG-treated group was significantly higher than that in the control group. However, the proportion of Jun showed an opposite trend and did not show any co-staining with HOPX (Figure 6E). However, in Fig. 6E, authors merely show pos. and neg. stained cells indicated by

arrows/arrowheads. In case the authors generated quantitative data, which would justify the use of significantly more or less, the author's should refer and present this information.

Response: The percentage of JUN positive cell decreased from 19.02%/19.66% (normal adult/iNOA) to 13.53%/12.82% (normal adult/iNOA) after ICG treatment and the difference was significant (Figure R3.10B). We will improve this maturation induction culture system in the further study to improve its efficiency. The quantification and its relative method has been added in the revised manuscript.

Figure R3.10. The change of JUN and HOPX in Sertoli cells after ICG treatment.

(A, B) Immunofluorescence co-staining of JUN (red) and HOPX (green) in cultured iNOA and normal adult Sertoli cells with or without ICG treatment (A). The statistics of JUN/HOPX positive cells count is shown as histogram (B). The scale bar represents 20 μ m. * p <0.05, ** p <0.01, *** p <0.001 (compared with DMSO treated group).

In general, this manuscript would greatly benefit from a summarizing model, which should cover the normal developmental changes of Sertoli cells plus the alterations in the patient groups that have been analyzed.

Response: Thank you for your suggestion, we have added a schematic diagram as the Figure 7 in the revised manuscript. It includes the 3-stage model of normal Sertoli cells development and the alterations in iNOA, KS and AZFa_Del patients.

Figure R3.11. The schematic diagram of this study showed the normal developmental changes of Sertoli cells and their alterations in the three types of NOA patients.

Reference

- Bergmann, M.W. (2010). WNT signaling in adult cardiac hypertrophy and remodeling: lessons learned from cardiac development. *Circulation research* 107, 1198-1208.
- Carlsen, E., Olsson, C., Petersen, J.H., Andersson, A.M., and Skakkebaek, N.E. (1999). Diurnal rhythm in serum levels of inhibin B in normal men: relation to testicular steroids and gonadotropins. *J Clin Endocrinol Metab* 84, 1664-1669.
- Cui, Y., Wu, X., Lin, C., Zhang, X., Ye, L., Ren, L., Chen, M., Yang, M., Li, Y., Li, M., *et al.* (2019). AKIP1 promotes early recurrence of hepatocellular carcinoma through activating the Wnt/β-catenin/CBP signaling pathway. *Oncogene* 38, 5516-5529.
- Esteves, S.C. (2015). Clinical management of infertile men with nonobstructive azoospermia. *Asian J Androl* 17, 459-470.
- Goluža, T., Boscanin, A., Cvetko, J., Kozina, V., Kosović, M., Bernat, M.M., Kasum, M.,

Kaštelan, Z., and Ježek, D. (2014). Macrophages and Leydig cells in testicular biopsies of azoospermic men. *BioMed research international* 2014, 828697.

Guo, J., Nie, X., Giebler, M., Mlcochova, H., Wang, Y., Grow, E.J., DonorConnect, Kim, R., Tharmalingam, M., Matilionyte, G., *et al.* (2020). The Dynamic Transcriptional Cell Atlas of Testis Development during Human Puberty. *Cell Stem Cell*.

Hall, A. (1998). Rho GTPases and the actin cytoskeleton. *Science (New York, NY)* 279, 509-514.

Hess, R.A., and França, L.R. (2005). Chapter 3 - Structure of the Sertoli Cell. In *Sertoli Cell Biology*, M.K. Skinner, and M.D. Griswold, eds. (San Diego: Academic Press), pp. 19-40.

Jiang, L., Yin, M., Wei, X., Liu, J., Wang, X., Niu, C., Kang, X., Xu, J., Zhou, Z., Sun, S., *et al.* (2015). Bach1 Represses Wnt/ β -Catenin Signaling and Angiogenesis. *Circulation research* 117, 364-375.

Krausz, C., Hoefsloot, L., Simoni, M., Tuttelmann, F., European Academy of, A., and European Molecular Genetics Quality, N. (2014). EAA/EMQN best practice guidelines for molecular diagnosis of Y-chromosomal microdeletions: state-of-the-art 2013. *Andrology* 2, 5-19.

Möller-Levet, C.S., Archer, S.N., Bucca, G., Laing, E.E., Slak, A., Kabiljo, R., Lo, J.C., Santhi, N., von Schantz, M., Smith, C.P., *et al.* (2013). Effects of insufficient sleep on circadian rhythmicity and expression amplitude of the human blood transcriptome. *Proceedings of the National Academy of Sciences of the United States of America* 110, E1132-1141.

Schneider, M.D., Baker, A.H., and Riley, P. (2015). Hopx and the Cardiomyocyte Parentage. *Molecular therapy : the journal of the American Society of Gene Therapy* 23, 1420-1422.

Shami, A.N., Zheng, X., Munyoki, S.K., Ma, Q., Manske, G.L., Green, C.D., Sukhwani, M., Orwig, K.E., Li, J.Z., and Hammoud, S.S. (2020). Single-Cell RNA Sequencing of Human, Macaque, and Mouse Testes Uncovers Conserved and Divergent Features of Mammalian Spermatogenesis. *Dev Cell*.

Sohni, A., Tan, K., Song, H.W., Burow, D., de Rooij, D.G., Laurent, L., Hsieh, T.C., Rabah, R., Hammoud, S.S., Vicini, E., *et al.* (2019). The Neonatal and Adult Human Testis Defined at the Single-Cell Level. *Cell Rep* 26, 1501-1517 e1504.

Steger, K., Rey, R., Louis, F., Kliesch, S., Behre, H.M., Nieschlag, E., Hoepffner, W., Bailey, D., Marks, A., and Bergmann, M. (1999). Reversion of the differentiated phenotype and maturation block in Sertoli cells in pathological human testis. *Hum Reprod* 14, 136-143.

Tang, D., Liu, W., Li, G., He, X., Zhang, Z., Zhang, X., and Cao, Y. (2020). Normal fertility with deletion of sY84 and sY86 in AZFa region. *Andrology* 8, 332-336.

Tanwar, P.S., Kaneko-Tarui, T., Zhang, L., Rani, P., Taketo, M.M., and Teixeira, J. (2010). Constitutive WNT/beta-catenin signaling in murine Sertoli cells disrupts their differentiation and ability to support spermatogenesis. *Biol Reprod* 82, 422-432.

Wang, M., Liu, X., Chang, G., Chen, Y., An, G., Yan, L., Gao, S., Xu, Y., Cui, Y., Dong, J., *et al.* (2018). Single-Cell RNA Sequencing Analysis Reveals Sequential Cell Fate Transition during Human Spermatogenesis. *Cell Stem Cell* 23, 599-614 e594.

Yu, W., Li, L., Zheng, F., Yang, W., Zhao, S., Tian, C., Yin, W., Chen, Y., Guo, W., Zou, L., *et al.* (2017). β -Catenin Cooperates with CREB Binding Protein to Promote the Growth of Tumor Cells. *Cellular physiology and biochemistry : international journal of experimental cellular physiology, biochemistry, and pharmacology* 44, 467-478.

Reviewers' Comments:

Reviewer #1:

Remarks to the Author:

The authors have addressed all of my questions and concerns.

Reviewer #2:

Remarks to the Author:

The authors have made a considerable effort to address my concerns. The manuscript has been substantially improved and represents an important study for the field.

Minor comments:

1. The new description of the relevance of the GO term "rhythmic processes" for Sertoli cells (Lines 178-183) is quite speculative. I would suggest removing this additional text to improve flow of the manuscript.

2. When referring to identified markers of spermatogenic stages on line 111, the relevant publications should be cited.

Robin Hobbs

Reviewer #3:

Remarks to the Author:

Zhao et al. submitted a revised version of their manuscript ,Single-cell atlas of human developing and azoospermia patients' testicles reveals the roadmap and defects in somatic microenvironment.'

The authors have made a great effort to address all the points that have been raised and have further improved this highly interesting manuscript.

Few minor issues remain with regard to the aspects mentioned in the first round:

For the modified Figure S1C, the authors need to mention the normal reference values for T, FSH, LH and E2 in the manuscript.

The authors provide a very convincing staining for GATA4 (Fig. 3J/K), supporting the specificity of HOPX in the response letter. However, in the revised version of the manuscript only the merged image (GATA4/HOPX) is shown, which does not seem to convey the message as clearly. It may be advisable to include the GATA4 staining also in this figure.

There seems to be a spelling mistake in Figure 4 G: Fluorescence intensity (a.u.) in per Sertoli cell nucleus. Delete the 'in'?

With regard to Figure 6, the authors have provided IF-images supporting the presence of Sertoli cells in the cultures. If possible, the authors can provide a merged SOX9/SMA image to further strengthen this aspect.

REVIEWERS' COMMENTS

Reviewer #1 (Remarks to the Author):

The authors have addressed all of my questions and concerns.

Response: Thank you for your help to improve our manuscript!

Reviewer #2 (Remarks to the Author):

The authors have made a considerable effort to address my concerns. The manuscript has been substantially improved and represents an important study for the field.

Minor comments:

1. The new description of the relevance of the GO term "rhythmic processes" for Sertoli cells (Lines 178-183) is quite speculative. I would suggest removing this additional text to improve flow of the manuscript.

Response: Thank you for your advice, we have deleted this description in the revised manuscript.

2. When referring to identified markers of spermatogenic stages on line 111, the relevant publications should be cited.

Response: Thank you for your advice, we have cited relevant publications for each marker.

Robin Hobbs

Reviewer #3 (Remarks to the Author):

Zhao et al. submitted a revised version of their manuscript ,Single-cell atlas of human developing and azoospermia patients' testicles reveals the roadmap and defects in somatic microenvironment.'

The authors have made a great effort to address all the points that have been raised and have further improved this highly interesting manuscript.

Few minor issues remain with regard to the aspects mentioned in the first round:

For the modified Figure S1C, the authors need to mention the normal reference values for T, FSH, LH and E2 in the manuscript.

Response: thank you for your advice, we have added the normal reference values for T, FSH, LH and E2 in the right panel of Figure S

The authors provide a very convincing staining for GATA4 (Fig. 3J/K), supporting the specificity of HOPX in the response letter. However, in the revised version of the manuscript only the merged image (GATA4/HOPX) is shown, which does not seem to convey the message as clearly. It may be advisable to include the GATA4 staining also in this figure.

Response: thank you for your advice, we need to show the expression pattern change of two stage specific regulators with ages, if three pictures (with GATA4 staining) are shown in one age group, this section will be very crowded and not conducive to reading. GATA4 is used as a common Sertoli cell marker, it not the key result of this study, therefore, taking all things into consideration, we have not changed this part of legend. Thank you!

There seems to be a spelling mistake in Figure 4 G: Fluorescence intensity (a.u.) in per Sertoli cell nucleus. Delete the 'in'?

Response: Sorry for this mistake, we have revised it and other similar mistake, all the grammar and spelling of the text were also rechecked in the revised manuscript.

With regard to Figure 6, the authors have provided IF-images supporting the presence of Sertoli cells in the cultures. If possible, the authors can provide a merged SOX9/SMA image to further strengthen this aspect.

Response: Both of our current antibodies are rabbit-derived, we will provide a merged SOX9/SMA image to *protocolexchange* database in the next few months after purchasing mouse-derived antibodies, thank you!